# DiffVax: Optimization-Free Image Immunization Against Diffusion-Based Editing

**Tarik Can Ozden**\*
UIUC

**Ozgur Kara**\*
UIUC

**Oguzhan Akcin**
UT Austin

**Kerem Zaman**
UNC Chapel Hill

**Shashank Srivastava**
UNC Chapel Hill

**Sandeep P. Chinchali**
UT Austin

**James M. Rehg**
UIUC

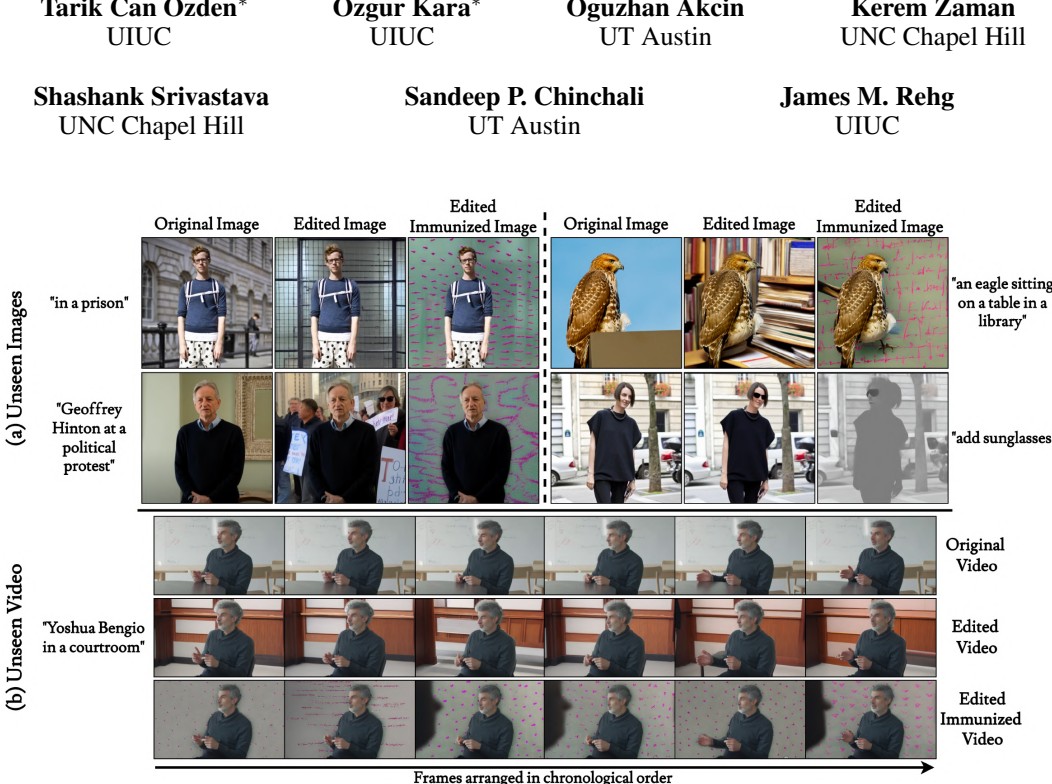

Figure 1: `DiffVax` is an optimization-free image immunization approach designed to protect images and videos from diffusion-based editing. `DiffVax` demonstrates robustness across diverse content, providing protection for both in-the-wild (a) *unseen images* and (b) *unseen video* content while effectively preventing edits across various editing methods, including *inpainting* (illustrated with a *human* in the left column and a *non-human foreground object* in the right column) and *instruction-based edits* (right column) with InstructPix2Pix (Brooks et al., 2023).

## Abstract

Current image immunization defense techniques against diffusion-based editing embed imperceptible noise into target images to disrupt editing models. However, these methods face scalability challenges, as they require time-consuming optimization for each image separately, taking hours for small batches. To address these challenges, we introduce `DiffVax`, a scalable, lightweight, and optimization-free framework for image immunization, specifically designed to prevent diffusion-based editing. Our approach enables effective generalization to unseen content, reducing computational costs and cutting immunization time from days to milliseconds, achieving a speedup of 250,000×. This is achieved through a loss term that ensures the failure of editing attempts and the imperceptibility of the perturbations. Extensive qualitative and quantitative results demonstrate that our model is scalable, optimization-free, adaptable to various diffusion-based editing tools, robust against counter-attacks, and, for the first time, effectively protects video content from editing. More details are available in our Project Webpage.

---

\*Equal contribution

# 1 INTRODUCTION

Recent advancements in generative models, particularly diffusion models (Sohl-Dickstein et al., 2015; Ho et al., 2020; Rombach et al., 2022), have enabled realistic content synthesis, which can be used for various applications, such as image generation (Saharia et al., 2022; Ruiz et al., 2023; Chefer et al., 2023; Zhang et al., 2023c; Li et al., 2023b; Mou et al., 2024b; Bansal et al., 2023) and editing (Brooks et al., 2023; Couairon et al., 2023a; Hertz et al., 2023b; Meng et al., 2022). However, the widespread availability and accessibility of these models introduce significant risks, as malicious actors exploit them to produce deceptive, realistic content known as deepfakes (Pei et al., 2024). Deepfakes pose severe threats across multiple domains, from political manipulation (Appel & Prietzel, 2022) and blackmail (Blancaflor et al., 2024) to biometric fraud (Wojewidka, 2020) and compromising trust in legal processes (Delfino, 2022). Furthermore, they have become tools for sexual harassment through the creation of non-consensual explicit content (Jean Mackenzie, 2024; Davies & McDermott, 2022; Cole, 2018). Given the widespread accessibility of diffusion models, the scale of these threats continues to grow, underscoring the urgent need for robust defense mechanisms to protect individuals, institutions, and public trust from such misuse.

To address these challenges, one line of research has focused on deepfake detection (Naitali et al., 2023; Passos et al., 2024) and verification methods (Hasan & Salah, 2019), which facilitate post-hoc identification. While effective for detection, these approaches do not proactively prevent malicious editing, as they only identify it after it happens. Another branch modifies the parameters of editing models (Li et al., 2024) to prevent unethical content synthesis (e.g. NSFW material); however, the widespread availability of unrestricted generative models limits its effectiveness. A more robust defense mechanism, known as image immunization (Salman et al., 2023; Lo et al., 2024; Yeh et al., 2021; Ruiz et al., 2020), safeguards images from malicious edits by embedding imperceptible adversarial perturbation. This approach ensures that any editing attempts lead to unintended or distorted results, proactively preventing malicious modifications rather than depending on post-hoc detection. The subtlety of this protection is particularly valuable for large-scale, publicly accessible content, such as social media, where user data is especially vulnerable to malicious attacks. By uploading immunized images instead of the original ones, users can reduce the risk of misuse by malicious actors, highlighting the potential of immunization-based methods for real-world impact.

However, current immunization approaches remain inadequate, as they do not simultaneously satisfy the key requirements of an effective defense: (i) scalability for large-scale content, (ii) memory and runtime efficiency, and (iii) robustness against counter-attacks. PhotoGuard (Salman et al., 2023) (PG) embeds adversarial perturbations into target images to disrupt components of the diffusion model by solving a constrained optimization problem via projected gradient descent (Madry et al., 2018a). Although PhotoGuard was the first immunization model targeting diffusion-based editing, it requires over 10 minutes of runtime per image and at least 15GB of memory, causing both computational and time inefficiency. To alleviate these demands, DAYN (Lo et al., 2024) proposes a semantic-based attack that disrupts the diffusion model's attention mechanism during editing. While this approach reduces computational load, it remains time-inefficient like PhotoGuard, as it requires a separate optimization process for each image and cannot generalize to unseen content. Furthermore, both approaches are vulnerable to counter-attacks, such as denoising the added perturbation or applying JPEG compression (Sandoval-Segura et al., 2023) to the immunized image. Consequently, neither method is practical for large-scale applications, such as safeguarding the vast volume of image and video data uploaded daily on social media platforms.

To address these challenges, we introduce `DiffVax`, an end-to-end framework for training an "immunizer model" that learns how to generate imperceptible perturbations to immunize target images against diffusion-based editing (see Fig 2). This immunization process ensures that any attempt to edit the immunized image using a diffusion-based model fails. `DiffVax` is more effective than prior works in ensuring editing failure, and it demonstrates the feasibility and generalizability of the image-conditioned feed-forward approach to perturbation generation.

Our training process is guided by two objectives, expressed as separate terms in the loss function: (1) encouraging the model to generate an imperceptible perturbation, and (2) ensuring that any editing attempt on the immunized image fails. Our trained immunizer operates with a single forward pass, completed within milliseconds, eliminating the need for time-intensive per-image optimization. This efficiency enables scalability to high-volume content protection. Additionally, `DiffVax` enhances

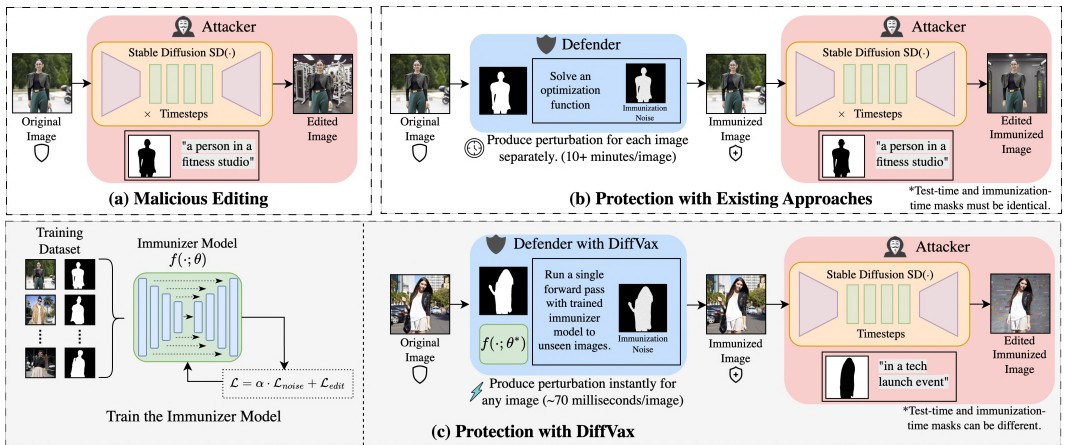

Figure 2: *Comparing* `DiffVax` *with existing approaches.* **(a)** An attacker performs malicious editing on an original image. **(b)** Existing defenses immunize images by solving a costly optimization problem for each image individually, taking over 10 minutes per image. **(c)** `DiffVax` enables scalable protection by first training an immunizer model (green box) on a diverse dataset. Once trained, the model can immunize unseen images with a single forward pass, producing effective perturbations in approximately 70 milliseconds per image.

memory efficiency by avoiding gradient computation during inference, setting it apart from prior methods. It also exhibits robustness against common counter-attacks, such as JPEG compression and image denoising (Sandoval-Segura et al., 2023). In addition, our framework demonstrates superior generalization with other diffusion-based editing methods (see Fig. 1 for examples on inpainting and instruction-based editing). Leveraging these strengths, we extend immunization to video content for the first time, achieving results previously unattainable due to the computational limitations of earlier approaches. As a result, `DiffVax` satisfies all key requirements for an effective defense.

To summarize, our contributions are as follows:

- We are the first to introduce a training framework in which the model learns to effectively immunize a given image against diffusion-based editing, drastically reducing inference time from days to milliseconds and enabling real-time protection.
- Thanks to its computational efficiency, our model shows promising potential as a foundational step toward immunizing video content.
- Unlike prior methods that require per-image optimization and therefore cannot generalize to unseen data, our approach enables generalization to new content through a learned "image immunizer".
- `DiffVax` achieves superior results with substantial degradation of the editing operation, and minimal memory requirement, demonstrating resistance to counter-attacks, making it the fastest, most cost-effective, and robust method available.

## 2 RELATED WORK

**Adversarial attacks**  Adversarial attacks exploit model vulnerabilities by introducing perturbations that induce misclassification. Early gradient-based methods efficiently generated such examples via gradient manipulation (Goodfellow et al., 2015; Madry et al., 2018b), later refined to minimize perceptual distortion (Carlini & Wagner, 2017; Moosavi-Dezfooli et al., 2016). Generative approaches advanced these attacks by synthesizing realistic adversarial inputs (Xiao et al., 2018). Subsequent work improved transferability and query efficiency using momentum and random search (Dong et al., 2018; Andriushchenko et al., 2020), while ensemble-based methods strengthened robustness evaluation (Croce & Hein, 2020). Universal perturbations (Moosavi-Dezfooli et al., 2017; Hayes & Danezis, 2018) and generative perturbation networks (Poursaeed et al., 2018) further generalized attacks across data and models. Building on these advances, our work focuses on immunizing against diffusion-based editing, addressing its unique characteristics.

**Preventing image editing** The proliferation of Latent Diffusion Models (LDMs) has underscored the demand for robust immunization strategies against unauthorized image manipulation. Initial efforts focused on Generative Adversarial Network (GAN)-based models, employing adversarial perturbations to inhibit edits (Yeh et al., 2021; Aneja et al., 2022). PhotoGuard (Salman et al., 2023) extended this line of work to diffusion models via encoder- and model-level perturbations but incurred substantial computational overhead due to backpropagation across multiple timesteps. To alleviate this, Lo et al. (2024)[1] proposed an attention-disruption strategy that bypasses full gradient computation, though its reliance on fixed prompts limits robustness. DiffusionGuard (Choi et al., 2025) enhances PhotoGuard by optimizing over augmented masks, yet remains computationally intensive. Similarly, PCA (Guo et al., 2024) proposes a grey-box attack by inducing posterior collapse in the VAE encoder to disrupt editing. Addressing instruction-guided editing, EditShield (Chen et al., 2023) introduces perturbations to shift latent representations, causing semantic mismatches in the edited output. Meanwhile, Shih et al. (2025) bypass the reliance on VAE encoders by proposing a feature-based attack effective against pixel-domain diffusion models. Other approaches, including Mist (Liang & Wu, 2023), AdvDM (Liang et al., 2023), SDS (Xue et al., 2024), and Glaze (Shan et al., 2023), target text-to-image diffusion or fine-tuned models, but exhibit high computational demands and limited resilience to adaptive attacks. In contrast, `DiffVax` introduces a model-agnostic immunizer that generalizes to unseen data via a single forward pass. Furthermore, we present, for the first time, promising results in the direction of video immunization.

**Diffusion-based image editing** Diffusion models have emerged as powerful tools for image editing tasks such as inpainting (Wang et al., 2023; Lugmayr et al., 2022; Zhang et al., 2023a), style transfer (Wang et al., 2023; Mou et al., 2024a; Yang et al., 2023; Hertz et al., 2023a), and text-guided transformations (Brooks et al., 2023; Lin et al., 2024; Ravi et al., 2023), by conditioning on prompts or image regions. Edits are guided through attention manipulation (Parmar et al., 2023) and multi-step noise prediction. Approaches include both training-based (Couairon et al., 2023b; Kim et al., 2022) and training-free methods (Mokady et al., 2023; Miyake et al., 2023) requiring minimal fine-tuning. We use stable diffusion inpainting as our primary editing model and include results with InstructPix2Pix (Brooks et al., 2023) to show model-agnostic performance.

## 3 METHODOLOGY

### 3.1 PRELIMINARIES

**Image immunization** Adversarial attacks exploit the vulnerabilities of machine learning models by introducing small, imperceptible perturbations to input data, causing the model to produce incorrect or unintended outputs (Szegedy et al., 2014; Biggio et al., 2013). In the context of diffusion models, such perturbations can be crafted to disrupt the editing process, ensuring that attempts to modify an adversarially perturbed image fail to achieve intended outcomes. Given an image $\mathbf{I}$, the goal is to transform it into an adversarially immunized version, $\mathbf{I}_{\mathrm{im}}$, by introducing a perturbation $\epsilon_{\mathrm{im}}$:

$$\mathbf{I}_{\mathrm{im}} = \mathbf{I} + \epsilon_{\mathrm{im}}, \quad \text{subject to:} \quad \|\epsilon_{\mathrm{im}}\|_p < \kappa, \tag{1}$$

where $\kappa$ is the perturbation budget that constrains the norm of the perturbation to ensure that it remains imperceptible. The norm $p$ could be chosen as 1, 2, or $\infty$, depending on the application.

**Latent diffusion models** LDMs (Rombach et al., 2022) perform the generative process in a lower-dimensional latent space rather than pixel space, achieving computational efficiency while maintaining high-quality outputs. This design is ideal for large-scale tasks like image editing and inpainting. Training an LDM starts by encoding the input image $\mathbf{I}_0$ into a latent representation $z_0 = \mathcal{E}(\mathbf{I}_0)$ using encoder $\mathcal{E}(\cdot)$. The diffusion process operates in this latent space, adding noise over $T$ steps to generate a sequence $z_1, \ldots, z_T$, with $z_{t+1} = \sqrt{1 - \beta_t} \, z_t + \sqrt{\beta_t} \, \epsilon_t$, $\epsilon_t \sim \mathcal{N}(\mathbf{0}, \mathbf{I})$, where $\beta_t$ is the noise schedule at step $t$. The training aims to learn a denoising network $\epsilon_\theta$ that predicts the added noise $\epsilon_t$ by minimizing $\mathcal{L}(\theta) = \mathbb{E}_{t, z_0, \epsilon \sim \mathcal{N}(\mathbf{0}, \mathbf{I})} \left[ \|\epsilon - \epsilon_\theta(z_t, t)\|_2^2 \right]$. In the reverse process, a noisy latent vector $z_T \sim \mathcal{N}(\mathbf{0}, \mathbf{I})$ is iteratively denoised via the trained denoising network to recover $z_0$, which is decoded into the final image $\tilde{\mathbf{I}} = \mathcal{D}(z_0)$ with decoder $\mathcal{D}(\cdot)$.

---

[1]Code unavailable despite request.

Figure 3: ***Overview of DiffVax.*** Our end-to-end training framework is illustrated in (a). The training process consists of two stages. In Stage 1, immunization is applied to the training image $\mathbf{I}$. In Stage 2, the immunized image $\mathbf{I}_{\mathrm{im}}$ is edited using a stable diffusion model $\mathrm{SD}(\cdot)$ with the specified text prompt and mask, during which the $\mathcal{L}_{\mathrm{noise}}$ and $\mathcal{L}_{\mathrm{edit}}$ are computed. During inference (b), the trained immunizer model generates immunization noise (see Inference Stage 1 in (b)) applied to the original (target) image using an immunization mask. When a malicious user attempts to attack these immunized images with an editing mask, the editing tool (see Inference Stage 2 in (b)) is unable to produce the intended edited content.

## 3.2 PROBLEM FORMULATION

Let $\mathbf{I} \in \mathbb{R}^{H \times W \times C}$ represent an image with height $H$, width $W$, and $C$ color channels. A malicious user using a diffusion-based editing tool, $\mathrm{SD}(\cdot)$, attempts to maliciously edit the image based on a prompt $\mathcal{P}$ and a binary mask $\mathbf{M} \in \{0, 1\}^{H \times W \times C}$, which defines the target area for editing, with a value of 1 indicating the region of interest and 0 denotes the background or irrelevant areas. Ideally, this target region can represent any meaningful part of the image, such as a human body or a face. Our objective is to immunize the original (target) image $\mathbf{I}$ by carefully producing a noise $\epsilon_{\mathrm{im}}$ that satisfies two key criteria: (a) $\epsilon_{\mathrm{im}}$ remains imperceptible to the user, and (b) the edited immunized image $\mathbf{I}_{\mathrm{im,edit}}$ fails to accurately reflect the prompt $\mathcal{P}$ applied by the malicious users. In other words, the immunized image disrupts the editing model $\mathrm{SD}(\cdot)$ such that any attempt to edit the image results in unsuccessful or unintended modifications. While our approach is broadly applicable to any diffusion-based editing tool, such as inpainting models and InstructPix2Pix (Brooks et al., 2023), this study follows previous work (Salman et al., 2023; Lo et al., 2024) by using inpainting as the primary editing tool for problem formulation and quantitative experiments. We focus on scenarios where the sensitive regions such as human body or face remains constant, with other areas considered editable, reflecting real-world malicious editing scenarios. Additional results for other objects and tools (e.g. InstructPix2Pix) are provided in Fig. 1, Fig. 4, and in our Appendix A.2.

## 3.3 OUR APPROACH

**End-to-end training framework** To overcome the speed limitations of previous methods, which require solving an optimization problem independently for each image, we propose an end-to-end training framework. This framework enables an immunizer model $f(\cdot; \theta)$ to instantly generate immunization noise for a given input image. Our training algorithm (see Appendix A.1, and Fig. 3 (a)) consists of two stages. In the first stage, we employ a UNet++ (Zhou et al., 2018) architecture for the "immunizer" model $f(\cdot; \theta)$, which takes an input image $\mathbf{I}$ and generates the corresponding immunization noise $\epsilon_{\mathrm{im}}$. Subsequently, $\epsilon_{\mathrm{im}}$ is multiplied by the immunization mask $\mathbf{M}$, which targets the region of interest (e.g. a person's face). The resulting masked noise is then added to the training image to produce the immunized image, computed as $\mathbf{I}_{\mathrm{im}} = \mathbf{I} + \epsilon_{\mathrm{im}} \odot \mathbf{M}$. Finally, the image is clamped to the $[0, 1]$ range. To ensure the noise remains imperceptible to the human eye, we introduce the following loss:

$$\mathcal{L}_{\mathrm{noise}} = \frac{1}{\mathrm{sum}(\mathbf{M})} \|(\mathbf{I}_{\mathrm{im}} - \mathbf{I}) \odot \mathbf{M}\|_p \tag{2}$$

where $p$ is empirically chosen to be 1. $\mathcal{L}_{\mathrm{noise}}$ penalizes deviations within the masked region, ensuring that the change between the immunized image and the training image is imperceptible. In the second stage, after generating the immunized image $\mathbf{I}_{\mathrm{im}}$, we apply diffusion-based editing using the editing tool $\mathrm{SD}(\cdot)$. This model takes the immunized image $\mathbf{I}_{\mathrm{im}}$, the training mask $\mathbf{M}$, and the training prompt $\mathcal{P}$ as input, performing edits in the regions specified by the mask. To ensure that the edited image is effectively distorted, we define the loss function:

$$\mathcal{L}_{\mathrm{edit}} = \frac{1}{\mathrm{sum}(\sim \mathbf{M})} \|\mathrm{SD}(\mathbf{I}_{\mathrm{im}}, \sim \mathbf{M}, \mathcal{P}) \odot (\sim \mathbf{M})\|_1, \tag{3}$$

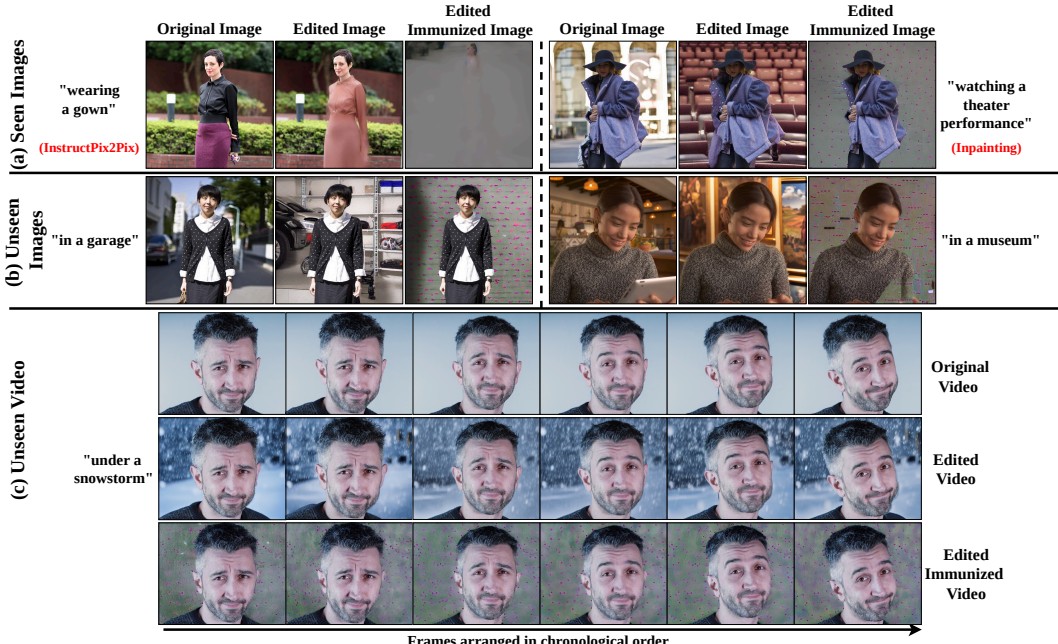

Figure 4: *Qualitative results with* `DiffVax`*.* Our method effectively immunizes (a) seen images and generalizes to (b) unseen images with diverse text prompts. Additionally, it extends to (c) unseen human videos, demonstrating its adaptability to new content. Furthermore, it supports various poses and perspectives, from full-body shots (a) to close-up face shots (c).

where $\sim \mathbf{M}$ represents the complement of the masked area and $\mathrm{SD}(\cdot)$ is the stable diffusion inpainting model that modifies the region $\sim \mathbf{M}$ in $\mathbf{I}_{\mathrm{im}}$ according to the prompt $\mathcal{P}$. This loss function is the key to our method, as it ensures that the immunization noise disrupts the editing process by forcing the unmasked regions to be filled with 0s. Note that for editing models that do not rely on masks, we exclude masks from the loss calculations.

To enable training, we curate a dataset of image, mask, and prompt tuples, represented as $\mathcal{D} = \{(\mathbf{I}^k, \mathbf{M}^k, \mathcal{P}^k)\}_{k=1}^N$. Specifically, we collect 1000 images of individuals from the CCP (Yang et al., 2014) dataset and use the Segment Anything Model (SAM) (Kirillov et al., 2023) to generate masks corresponding to the foreground objects in these images. To ensure diverse text descriptions for the editing tasks, we utilize ChatGPT OpenAI (2024) (see Appendix A.1). At each training step, a sample is selected from the dataset and initially processed by the immunizer model $f(\cdot; \theta)$ to generate immunization noise $\epsilon_{\mathrm{im}}^n$, which is added to the masked region of the training image and then clamped. The resulting immunized image $\mathbf{I}_{\mathrm{im}}^n$ is then passed through the editing model $\mathrm{SD}(\cdot)$ to produce the edited immunized image $\mathbf{I}_{\mathrm{im,edit}}^n$. The final loss function, $\mathcal{L} = \alpha \cdot \mathcal{L}_{\mathrm{noise}} + \mathcal{L}_{\mathrm{edit}}$, is used for backpropagation with respect to the immunizer model's parameters. Backpropagating through the stable diffusion stages allows the immunizer to learn the interaction between the perturbation and the generated pixels. Through this iterative process, the immunizer model learns to generate perturbations that disrupt the editing model. Following the insights from PhotoGuard's encoder attack, we do not condition the immunizer model on text prompts, as the noise is empirically shown to be prompt-agnostic (see Appendix A.6).

**Inference** During inference, the trained immunizer model generates immunization noise for any original (target) image using the mask of the region intended for protection. This noise is then applied to create the immunized image, with the noise restricted to the masked region. The resulting immunized image can be safely shared publicly. When a malicious user inputs this immunized image along with an editing mask into a diffusion-based editing tool (the same tool used during training), the immunization noise disrupts the edited output (see Fig. 3 (b)). Unlike previous approaches that require the same mask to be used during both training and inference, our method decouples these phases. This separation allows the immunizer model to generalize to unseen content, addressing the limitation of previous methods where malicious users could exploit different masks during editing (e.g. using an immunization mask of full-body but applying an editing mask of face).

## 4 EXPERIMENTATION

**Baselines** We compare `DiffVax` with several existing image immunization methods. As a naive baseline, we include **Random Noise**, which applies arbitrary noise to images. We also evaluate two variants of PhotoGuard (Salman et al., 2023): **PhotoGuard-E**, which embeds adversarial perturbations in the latent encoder, and **PhotoGuard-D**, which disrupts the entire generative process. Additionally, we compare against **DiffusionGuard** (Choi et al., 2025), an extension of PhotoGuard that augments masks during optimization. To evaluate robustness against counter-attacks, we develop three additional baselines where editing is applied after immunization: (i) passing the image through a convolutional neural network (CNN)-based denoiser (Li et al., 2023a), denoted as `DiffVax` w/ D.; (ii) compressing the image as JPEG (Sandoval-Segura et al., 2023) with a 0.75 compression ratio, denoted as `DiffVax` w/ JPEG; and (iii) applying the IMPRESS defense (Cao et al., 2023), denoted as `DiffVax` w/ IMPRESS.

**Evaluation metrics and dataset** We focus on four key aspects in evaluation: (a) *the amount of editing failure*, where we follow previous approaches (Salman et al., 2023) and utilize SSIM (Wang et al., 2004), PSNR and FSIM (Zhang et al., 2011) metrics to measure the visual differences between the edited immunized image and the edited original image; (b) *imperceptibility*, where the amount of the immunization noise quantified by measuring the SSIM between the original image and the immunized image, denoted as SSIM (Noise); (c) *the degree of textual misalignment* evaluated using CLIP (Radford et al., 2021) by measuring the average similarity between the edited immunized image and the text prompt, denoted as CLIP-T; and (d) *scalability* by reporting the average runtime and GPU memory required to immunize a single image on average from the dataset. We curate a dataset of 875 human images from the CCP (Yang et al., 2014) dataset. Of these, 800 images are used for training (including the 75 seen images in our experiments), and 75 unseen images are reserved for testing.

**Qualitative results** Figures 1 and 4 illustrate the qualitative success of our method. `DiffVax` effectively immunizes images against various editing techniques, including standard inpainting and instruction-based models like InstructPix2Pix (Brooks et al., 2023) (Figure 1). As further detailed in Appendix A.2, the model demonstrates a strong ability to generalize to unseen images and a wide range of prompts, accommodating various human perspectives from full-body to close-up shots (Figure 4). Although trained primarily on human subjects, our model also extends its robustness to non-human objects. When compared to baseline methods (Figure 5), our approach is qualitatively superior on both seen and unseen images, generating backgrounds that deviate more significantly from the intended edits. Notably, in many cases with our approach, it is impossible to infer the original prompt from the immunized image's background, a stark contrast to PhotoGuard, which often retains discernible hints of the prompt. More examples, including comparisons and results with other editing models, are provided in Appendix A.2.

**DiffVax is more effective in corrupting edits** As shown in Table 1, `DiffVax` achieves the lowest SSIM, PSNR, and FSIM values overall, securing second place in the SSIM metric for unseen data, with a small margin behind PG-D, indicating that malicious edits on immunized images are significantly distorted, even on previously unseen data, whereas baseline methods, which require optimization to be re-run for each image, do not differentiate between seen and unseen data. Additionally, CLIP-T results, which measure textual misalignment, further verify these findings by measuring the misalignment semantically in the edited immunized images. `DiffVax` outperforms the baselines by maintaining the highest SSIM (Noise) values for both seen and unseen data, highlighting its effectiveness in corrupting malicious edits while keeping the immunized image imperceptible. This superior imperceptibility is achieved because our model learns to generate visually subtle, low-frequency perturbations, in contrast to the scattered, high-frequency noise produced by prior methods (see Appendix A.5 for a detailed discussion). Thus, *training an immunizer model enables it to learn how to strategically place immunization noise to effectively disrupt diffusion-based editing, by aggregating over the training set.* In contrast, prior optimization-based works only see a single target image at a time.

**DiffVax is more scalable** In addition to its strong qualitative performance, `DiffVax` offers significant advantages in speed and memory efficiency. It completes the immunization process in just 0.07 seconds per image on average, compared to 207.0 seconds for PhotoGuard-E, 911.6 seconds for PhotoGuard-D, and 131.1 seconds for DiffusionGuard. In terms of GPU memory usage, `DiffVax` requires only 5,648 MiB, much lower than PhotoGuard-E (9,548 MiB), PhotoGuard-D (15,114

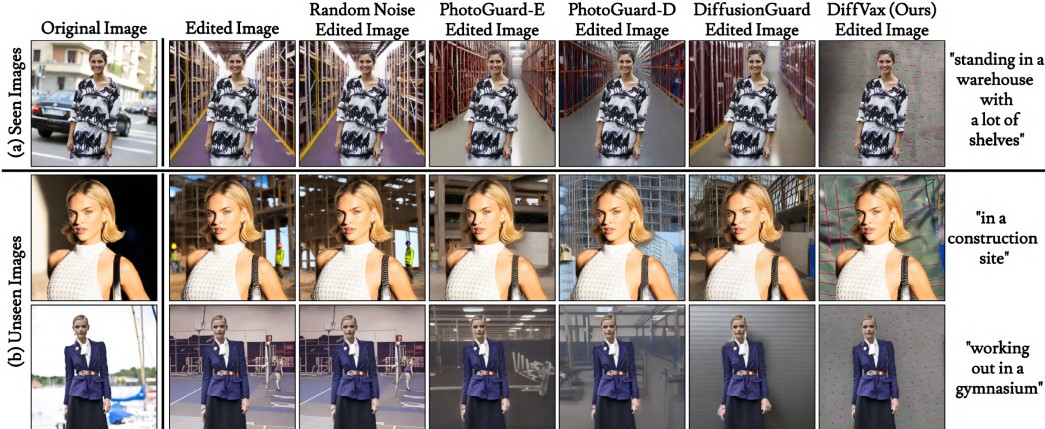

Figure 5: *Qualitative comparison of edited images across immunization methods.* This figure shows the results of different immunization methods: Random Noise, PhotoGuard-E, PhotoGuard-D, DiffusionGuard, and our proposed method, `DiffVax`. Results for (a) seen and (b) unseen images are shown, with different prompts applied to each (right side). The first column contains the original images, while subsequent columns show the edited outputs under different settings, as depicted on the top. Note that `DiffVax` is *substantially more effective* than PhotoGuard-E, -D and DiffusionGuard in degrading the edit.

Table 1: *Performance comparisons on images.* The SSIM, PSNR, FSIM, SSIM (Noise), and CLIP-T metrics are reported separately for the *seen* and *unseen* splits of the test dataset. Runtime and GPU requirements are measured as the average time (in seconds) and memory usage (in MiB) needed to immunize a single image. "N/A" indicates that the corresponding value is unavailable. The symbols ↑ and ↓ indicate the direction toward better performance for each metric, respectively. **Bold** values indicate the best scores, while underlined values denote the second-best scores.

| Immunization Method | Amount of Editing Failure | | | | | | Imperceptibility | | Text Misalignment | | Scalability | |
| | SSIM ↓ | | PSNR ↓ | | FSIM ↓ | | SSIM (Noise) ↑ | | CLIP-T ↓ | | Runtime (s) ↓ | GPU Req. (MiB) ↓ |
| | seen | unseen | seen | unseen | seen | unseen | seen | unseen | seen | unseen | (Immunization) | (Immunization) |
|---|---|---|---|---|---|---|---|---|---|---|---|---|
| Random Noise | 0.586 | 0.585 | 16.09 | 16.40 | 0.460 | 0.458 | 0.902 | 0.903 | 31.68 | 31.62 | N/A | N/A |
| PhotoGuard-E | 0.558 | 0.565 | 15.29 | 15.63 | 0.413 | 0.408 | 0.956 | 0.956 | 31.69 | 30.88 | 207.00 | 9,548 |
| PhotoGuard-D | 0.531 | **0.523** | 14.70 | 14.92 | 0.386 | 0.379 | 0.978 | 0.979 | 29.61 | 29.27 | 911.60 | 15,114 |
| DiffusionGuard | 0.551 | 0.556 | 14.37 | 14.71 | 0.389 | 0.386 | 0.965 | 0.966 | 26.98 | 27.10 | 131.10 | 6,750 |
| DiffVax (Ours) | **0.510** | 0.526 | **13.96** | **14.32** | **0.353** | **0.362** | **0.989** | **0.989** | **23.13** | **24.17** | **0.07** | **5,648** |

MiB), and DiffusionGuard (6,750 MiB). This makes *DiffVax a practical and scalable solution for large-scale applications.*

**DiffVax is more robust to counter-attacks** Table 2 shows that `DiffVax` is robust to common counter-attacks, including CNN-based denoising, JPEG compression, and IMPRESS (Cao et al., 2023). `DiffVax` consistently outperforms PhotoGuard-D across all scenarios, as further evidenced by the detailed results in Appendix A.3.3. This robustness arises from `DiffVax`'s ability to learn spatially targeted, low-frequency perturbations. Unlike existing approaches that produce more uniform, high-frequency noise, our method's perturbations are less susceptible to removal by techniques like JPEG compression, which discards high-frequency content, or by denoisers trained to suppress uniform noise. Crucially, as shown in Appendix A.3.2, `DiffVax` achieves superior edit disruption with a much smaller mean magnitude of noise than baselines with larger fixed budgets. This highlights that its strength lies in the strategic placement of noise, not simply its magnitude, supporting our claim that `DiffVax` learns a more efficient and targeted noise distribution. Furthermore, our extensive robustness evaluations in Appendix A.3 show that `DiffVax` also maintains its effectiveness against attackers who vary their inference-time settings, consistently outperforming baselines across different sampling steps and diffusion samplers.

**User study results** We also conduct a user study with 67 participants on Prolific (2024), in which participants compare the "unrealisticness" level of baselines, and the edited image across 20 randomly selected image pairs, including both seen and unseen samples. For each model, we report the average rank, with our model achieving the top position with an average rank of 1.64, demonstrating clear superiority (see Appendix A.3.5), followed by PhotoGuard-D with a rank of 2.63.

Table 2: *Performance comparisons on edits with counter-attacks.* We report the SSIM, SSIM (Noise) and CLIP-T metrics for the denoiser (D.), JPEG (compression ratio of 0.75) counter-attacks separately for the *seen* and *unseen* splits of the test dataset.

| Method | SSIM ↓ | | PSNR ↓ | | FSIM ↓ | | SSIM (Noise) ↑ | | CLIP-T ↓ | |
|---|---|---|---|---|---|---|---|---|---|---|
| | *seen* | *unseen* | *seen* | *unseen* | *seen* | *unseen* | *seen* | *unseen* | *seen* | *unseen* |
| PG-D w/ D. | 0.702 | 0.709 | 18.27 | 18.43 | 0.528 | 0.528 | **0.966** | **0.965** | 31.48 | 31.20 |
| DiffusionGuard w/ D. | 0.708 | 0.719 | 18.26 | 18.69 | 0.530 | 0.531 | 0.964 | 0.964 | 31.08 | 30.99 |
| `DiffVax` w/ D. | **0.552** | **0.565** | **14.48** | **14.91** | **0.388** | **0.392** | 0.960 | 0.960 | **27.32** | **27.74** |
| PG-D w/ JPEG | 0.664 | 0.674 | 17.32 | 17.68 | 0.495 | 0.501 | 0.956 | 0.956 | 32.15 | 32.48 |
| DiffusionGuard w/ JPEG | 0.680 | 0.684 | 17.45 | 17.83 | 0.505 | 0.503 | 0.951 | 0.951 | 31.52 | 31.53 |
| `DiffVax` w/ JPEG | **0.522** | **0.538** | **14.17** | **14.61** | **0.374** | **0.382** | **0.959** | **0.959** | **26.04** | **26.05** |
| PG-D w/ IMPRESS | 0.578 | 0.563 | 15.89 | 16.07 | 0.436 | 0.426 | 0.640 | 0.634 | 31.35 | 31.26 |
| DiffusionGuard w/ IMPRESS | 0.604 | 0.595 | 15.89 | 16.09 | 0.453 | 0.442 | 0.636 | 0.630 | 30.88 | 30.50 |
| `DiffVax` w/ IMPRESS | **0.488** | **0.500** | **14.04** | **14.38** | **0.355** | **0.359** | **0.644** | **0.637** | **24.88** | **25.27** |

Table 3: *Ablation study.* We report the SSIM and SSIM (Noise) metrics for each loss term ablation, with results presented individually for the seen and unseen splits of the dataset.

| Method | SSIM ↓ | | PSNR ↓ | | FSIM ↓ | | SSIM (Noise) ↑ | | CLIP-T ↓ | |
|---|---|---|---|---|---|---|---|---|---|---|
| | *s* | *u* | *s* | *u* | *s* | *u* | *s* | *u* | *s* | *u* |
| `DiffVax` w/o $\mathcal{L}_{\mathrm{noise}}$ | **0.508** | **0.520** | **13.57** | **13.82** | **0.335** | **0.344** | 0.785 | 0.786 | 24.34 | 25.78 |
| `DiffVax` w/o $\mathcal{L}_{\mathrm{edit}}$ | 0.944 | 0.932 | 31.36 | 31.05 | 0.821 | 0.806 | **0.999** | **0.999** | 32.01 | 32.27 |
| `DiffVax` | 0.510 | 0.526 | 13.96 | 14.32 | 0.353 | 0.362 | 0.989 | 0.989 | **23.13** | **24.17** |

**Ablation study** To assess the contribution of each component in our framework, we conduct an ablation study by individually removing $\mathcal{L}_{\mathrm{edit}}$ and $\mathcal{L}_{\mathrm{noise}}$. As shown in Table 3, when $\mathcal{L}_{\mathrm{noise}}$ is removed, the model achieves slightly better performance on unseen data in terms of failed immunized editing (measured by SSIM, PSNR, FSIM and CLIP-T). However, the immunization noise is no longer imperceptible, as indicated by the change in the SSIM (Noise) metric. Conversely, when $\mathcal{L}_{\mathrm{edit}}$ is removed, the SSIM (Noise) metric reaches its highest value, indicating minimal noise, but the model fails to prevent malicious editing, as reflected in the SSIM, PSNR, FSIM and CLIP-T metrics. Thus, *combining both terms in the final loss function is crucial for balancing imperceptibility and robustness in the training process* (see Appendix A.7).

## 5 CONCLUSION AND DISCUSSION

**Discussion on generalization** While a universal immunizer remains an open challenge, `DiffVax` demonstrates superior generalization over prior optimization-based work across three key dimensions. First, it addresses **generalization to unseen models**. Universal cross-model transferability is a difficult open problem, and like prior work, `DiffVax` is primarily model-specific and does not perfectly generalize to all unseen models. However, as detailed in Appendix A.4.1, it demonstrates significantly better performance in the challenging black-box transfer task from a model trained on Stable Diffusion (SD) v1.5 to an unseen SD v2 model. In this scenario, our learned immunization successfully transfers its protective effect, whereas optimization-based methods like PhotoGuard and DiffusionGuard fail completely, showing a clear improvement in cross-model robustness. Second, its feed-forward nature enables **generalization to unseen content**, a significant advantage over methods requiring costly per-image optimization. As discussed in Appendix A.4.2, the success of `DiffVax` proves that the set of effective perturbations has a learnable structure, allowing it to immunize new images, prompts, and even videos with a single pass. Finally, `DiffVax` is uniquely robust in its **generalization to unseen masks**. Unlike prior work, it does not overfit to the training mask's shape or scale, maintaining its edit-disrupting effectiveness even when test-time editing masks differ significantly from the immunization mask, as shown in Appendix A.4.3.

**Discussion on applications** To further validate the versatility of `DiffVax`, we evaluate its effectiveness across a broad spectrum of real-world editing scenarios, as illustrated in Figure 6. Our experiments demonstrate robust protection against diverse manipulation types, ranging from **global transformations** such as artistic style transfer (e.g., converting an image to Cubism) to **localized edits** like facial expression changes and object attribute modification. Furthermore, the framework proves effective in complex settings involving multiple object editing, object replacement, and tex-

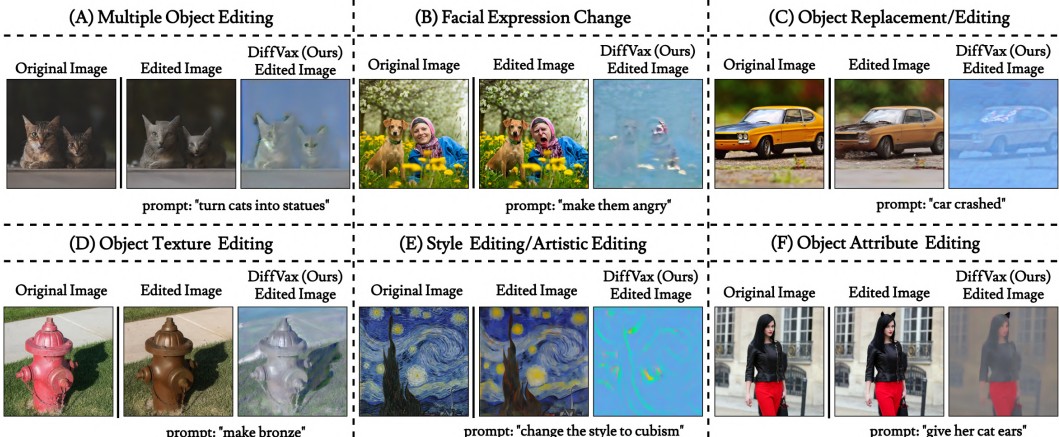

Figure 6: *DiffVax performance across diverse editing applications.* Our method effectively immunizes images against various manipulation types, including (A) multiple object editing, (B) facial expression changes, (C) object replacement, (D) texture editing, (E) artistic style transfer, and (F) attribute editing.

ture alteration. This wide applicability confirms that `DiffVax` does not merely memorize specific noise patterns for simple inpainting but learns to disrupt the underlying semantic guidance required for various high-level editing tasks.

**Discussion on editing models** Following prior work, our main evaluations are conducted using inpainting-based editing methods. However, we emphasize that our framework is model-agnostic and can be applied to various editing tools. To demonstrate this, we include additional results using the instruction-based model InstructPix2Pix (IP2P) (Brooks et al., 2023) (see Figure 10 in the Appendix) and the training-free model MagicBrush (Zhang et al., 2023b) (see Table 4 in the Appendix). We find that IP2P is particularly well-suited for complex or localized editing tasks, such as background modifications, stylistic changes, or edits outside sensitive regions, whereas inpainting-based approaches are more specialized for background editing tasks. Specifically, inpainting methods can introduce unintended alterations in sensitive areas like faces when the provided mask only partially covers the target region. This can conflict with the intent of a malicious user, whose goal is often to preserve identity while making selective edits.

**Conclusion** In this work, we present `DiffVax`, an optimization-free image immunization framework that protects against diffusion-based editing. Central to our approach is a trained "image immunizer" model that generates imperceptible perturbations to disrupt the editing process. At inference, `DiffVax` requires only a single forward pass, enabling scalability to large-scale deployments. Leveraging this efficiency, we extend our framework to video, demonstrating promising results for the first time (see Appendix A.3.6). Moreover, `DiffVax` is compatible with any diffusion-based editing tool and demonstrates strong robustness against counter-attacks. Overall, it establishes a new benchmark for scalable, real-time, and effective content protection.

ACKNOWLEDGMENTS

We acknowledge partial support from the Health Care Engineering Systems Center (Grainger College of Engineering, UIUC).

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
