# A APPENDIX

## CONTENTS

### A.1 MODEL ALGORITHM AND IMPLEMENTATION DETAILS

**Implementation Details** We employ a UNet++ architecture (Zhou et al., 2018) for the immunizer model. We selected this over a standard U-Net because its nested skip pathways facilitate denser feature aggregation at different semantic levels. Empirically, we found that this dense connectivity provides significantly better training stability for the unstable optimization task of predicting adversarial noise, allowing the model to generate more precise, imperceptible high-frequency perturbations. We train our immunizer model for 350 epochs using a batch size of 5 on an NVIDIA A100 GPU. We use the Adam optimizer (Kingma & Ba, 2015) with an initial learning rate of 0.00001 and set the loss weight parameter $\alpha = 4$. Training takes approximately 22 hours and leverages 16-bit precision to reduce memory consumption and speed up computation. As illustrated in Figure 7, the training process exhibits high stability; the noise loss converges rapidly to ensure imperceptibility, while the edit loss decreases steadily as the model progressively learns to disrupt the editing process. For the editing tools, we use a pre-trained Stable Diffusion v1.5 inpainting model (Rombach et al., 2022) for inpainting-based editing, and InstructPix2Pix (Brooks et al., 2023) for instruction-based editing tasks.

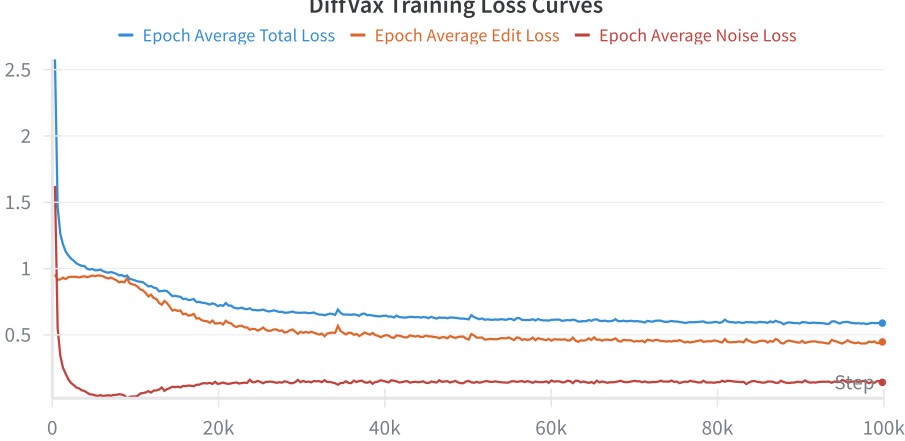

Figure 7: ***Training Loss Curves.*** We report the epoch-averaged Total Loss (blue), Edit Loss (orange), and Noise Loss (red). The curves demonstrate that the noise loss drops quickly to a low magnitude, ensuring the perturbation remains imperceptible, while the edit loss reduces continuously as the immunizer optimizes for edit disruption.

**Training Algorithm** Algorithm 1 describes the end-to-end training procedure for our immunizer model. For each data sample, the model generates an immunized image by injecting noise into the masked region. This image is then edited using a black-box editing model. The training objective minimizes both the deviation from the original image in the masked region and the effectiveness of the edit in the unmasked region.

---

**Algorithm 1** End-to-end Training Framework

---

**Input:** Immunizer model $f(\cdot; \theta)$, Editing model $\text{SD}(\cdot)$, Dataset $\mathcal{D}$, Dataset size $N$, Loss weight $\alpha$

    **for** $n = 1$ to $N$ **do**
        $(\mathbf{I}^n, \mathbf{M}^n, \mathcal{P}^n) \leftarrow \text{sample}(\mathcal{D}, n)$
        $\epsilon_{\text{im}}^n \leftarrow f(\mathbf{I}^n; \theta)$
        $\mathbf{I}_{\text{im}}^n \leftarrow (\mathbf{I}^n + \epsilon_{\text{im}}^n \odot \mathbf{M}^n).\text{clamp}(0, 1)$
        $\mathbf{I}_{\text{im,edit}}^n \leftarrow \text{SD}(\mathbf{I}_{\text{im}}^n, \sim \mathbf{M}^n, \mathcal{P}^n)$
        $\mathcal{L}_{\text{noise}} \leftarrow \text{normalize}(\|(\mathbf{I}_{\text{im}}^n - \mathbf{I}^n) \odot \mathbf{M}^n\|_1)$
        $\mathcal{L}_{\text{edit}} \leftarrow \text{normalize}(\|\mathbf{I}_{\text{im,edit}}^n \odot (\sim \mathbf{M}^n)\|_1)$
        $\mathcal{L} \leftarrow \alpha \cdot \mathcal{L}_{\text{noise}} + \mathcal{L}_{\text{edit}}$
        $\theta \leftarrow \text{update}(\nabla_\theta \mathcal{L})$
    **end for**

---

**Dataset Setup** Our dataset consists of 1,000 images, each associated with two prompts, resulting in a total of 2,000 prompts. We split the dataset into 80% for the training set (seen) and 20% for the validation set (unseen). The prompt set was constructed using ChatGPT (OpenAI, 2024), specifically by generating prompts designed for background editing. A total of 1,000 prompts were collected and subsequently split into 80% for the training set (seen) and 20% for the validation set (unseen). Finally, we sampled two random prompts for each image in the dataset, ensuring the prompts corresponded to whether the image was categorized as seen or unseen.

Our dataset is comparable in size to the current datasets used in related works, and is therefore aligned with the current standard of evidence in the field, while more data would always be better. To place our dataset size in the context of prior work, the closest research for training a generative adversarial noise generator is the paper "Generative Adversarial Perturbations" (Poursaeed et al., 2018). For their experiments on semantic segmentation, they used the focused Cityscapes dataset, which contains 2,975 training and 500 validation images. Given that this foundational work was established on a dataset of a few thousand images from a specific domain (urban scenes), we believe our dataset of 875 human images is in a comparable range for a proof-of-concept study. Nevertheless, we believe that extending our method to larger and more diverse datasets is a crucial next step, and we will highlight this as an important avenue for future work.

## A.2 ADDITIONAL QUALITATIVE RESULTS AND COMPARISONS

### A.2.1 ADDITIONAL RESULTS WITH INPAINTING-BASED EDITING MODELS

Figure 8 presents supplementary qualitative results obtained using inpainting-based editing models. The examples cover a wide range of scenarios and prompts, demonstrating the effectiveness of our immunization method on previously unseen content. Notably, the model performs well even on close-up images, maintaining robustness against malicious edits in both broad and fine-grained contexts.

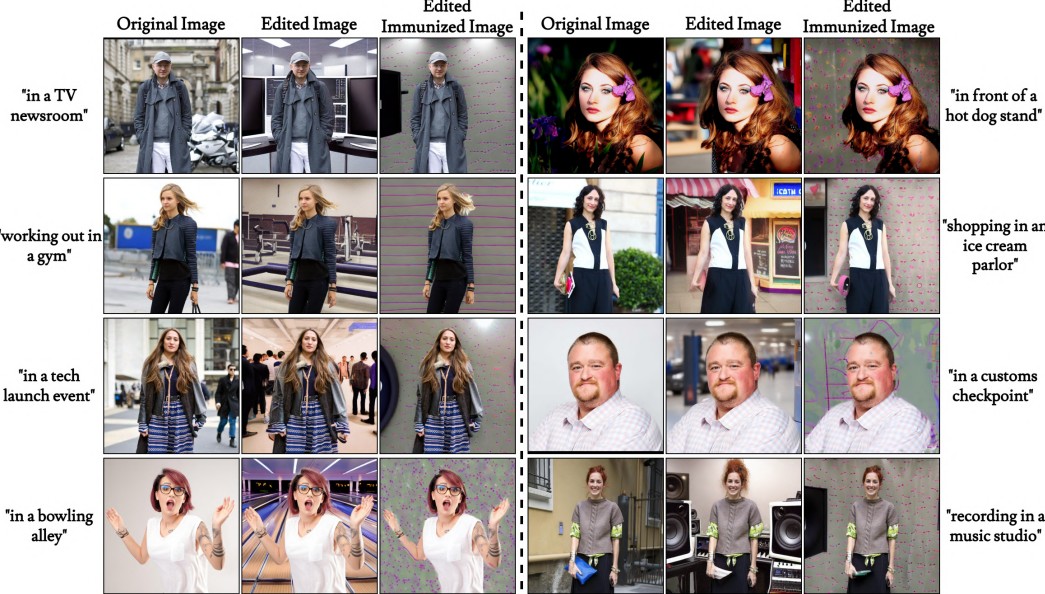

Figure 8: *Additional qualitative results with `DiffVax`.* Each row displays a different prompt and input image, illustrating `DiffVax`'s ability to consistently disrupt harmful edits. Despite varying and challenging prompts, the edited outputs from the protected images show clear signs of disruption, emphasizing the robustness of our method.

### A.2.2 ADDITIONAL COMPARISONS WITH INPAINTING-BASED EDITING MODELS

Figure 9 shows extended qualitative comparisons between DiffVax and various baseline immunization methods, including Random Noise, PhotoGuard-E, PhotoGuard-D, and DiffusionGuard. These results are produced using inpainting-based editing models. The comparison highlights how DiffVax consistently achieves better performance in visually disrupting malicious edits while preserving the semantic integrity of the original image.

We note that other defense methods such as AdvDM (Liang et al., 2023), SDS (Xue et al., 2024), and Mist (Liang & Wu, 2023) have also been proposed in the literature. However, these techniques are tailored for specific editing pipelines like SDEdit (Meng et al., 2022) and are not directly applicable in our inpainting-based setup, thus making direct comparison beyond our experimental scope.

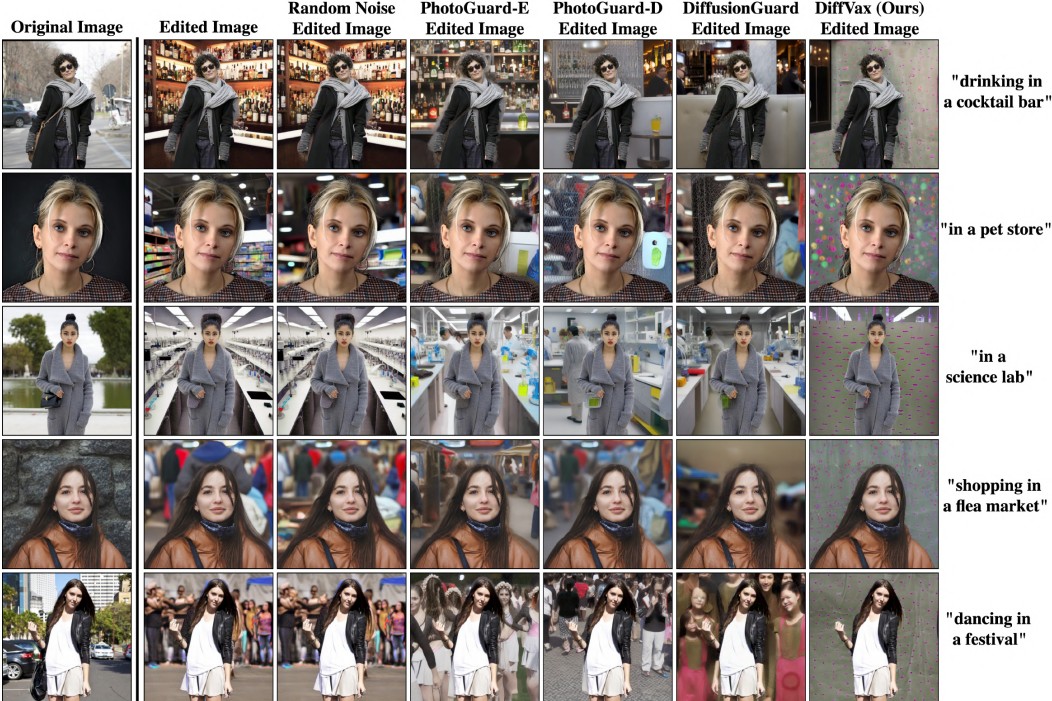

Figure 9: *Additional qualitative comparison between baselines and **DiffVax**.* Each row represents a unique prompt-image pair, while the columns show outputs for different immunization methods. DiffVax consistently produces better results, effectively disrupting edits while preserving image quality.

### A.2.3 ADDITIONAL RESULTS WITH INSTRUCTION-BASED EDITING MODEL

To further evaluate the generalizability of `DiffVax`, we apply it to edits generated using Instruct-Pix2Pix (Brooks et al., 2023), a widely adopted text-guided diffusion-based editing tool. This setting differs significantly from inpainting models, as edits are applied based on high-level natural language instructions. As shown in Figure 10, `DiffVax` consistently disrupts a broad range of editing intents across various image types. The examples illustrate the model's robustness across:

- **Human attribute edits** (e.g., *"add a hat to her head"*, *"add bowtie to person"*, *"make him wear a small scarf"*): `DiffVax` suppresses the addition of these features, effectively neutralizing changes to facial and clothing attributes.

- **Background edits** (e.g., *"make the background a chapel"*, *"change him to a statue"*): Despite significant changes to the scene, the edits fail to render properly on immunized images, showcasing `DiffVax`'s ability to neutralize edits in large non-focal areas.

- **Style transfer edits** (e.g., *"change the style to starry nights"*, *"make the style cubism"*, *"van gogh style"*): `DiffVax` prevents global transformations from taking effect, demonstrating its efficacy in blocking even abstract stylistic alterations.

- **Non-ROI edits** (e.g., *"add hot-air balloons to back"*, *"add necklace to person"*, *"add headphones"*): These involve subtle object insertions in the background or around the subject. Even though the modification targets are not directly in the immunized region, `DiffVax` still effectively disrupts the edit.

These results validate the model-agnostic and instruction-resilient nature of `DiffVax`, confirming its applicability to both local and global edit intents.

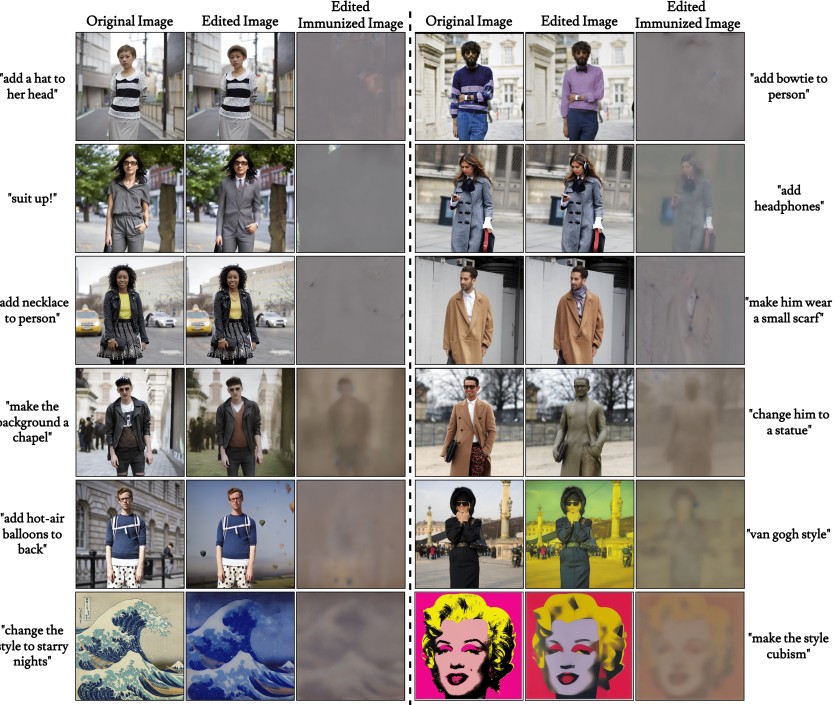

Figure 10: ***Qualitative results using the InstructPix2Pix (Brooks et al., 2023) editing model with DiffVax.*** Each triplet shows an original image, its edited counterpart, and the result after immunization. `DiffVax` successfully prevents a diverse set of edits, including background replacement, style transfer, object insertion, and attribute modification, further demonstrating its generalizability across editing types.

### A.2.4 Additional Evaluation with MagicBrush and Other Editing Models

The landscape of generative editing models is vast and rapidly evolving. Our choice of evaluation models was guided by established benchmarks in the image immunization literature to ensure a fair and direct comparison with prior state-of-the-art methods. To further strengthen our claims of generalizability, we conducted an additional experiment comparing our approach against PhotoGuard on the modern, training-free editing model MagicBrush. As shown in Table 4, our learned perturbations remain effective at disrupting edits, demonstrating that the protection generalizes beyond the standard inpainting and instruction-based models used in prior benchmarks. Our preliminary results show that `DiffVax` achieves superior edit disruption (lower SSIM, PSNR, FSIM, and CLIP-T) with comparable imperceptibility (SSIM Noise).

Table 4: *MagicBrush Comparison*

| **MagicBrush** | SSIM $\downarrow$ | PSNR $\downarrow$ | FSIM $\downarrow$ | CLIP-T $\downarrow$ | SSIM (Noise) $\uparrow$ |
|---|---|---|---|---|---|
| PhotoGuard | 0.682 | 18.81 | 0.546 | 25.64 | **0.967** |
| DiffVax | **0.635** | **18.41** | **0.529** | **22.18** | 0.965 |

Table 5 contextualizes our evaluation scope by comparing the editing tools used across recent immunization works. The scope of our evaluation is aligned with current best practices. We acknowledge that methods like Prompt-to-Prompt Hertz et al. (2023b) and Null-text inversion Mokady et al. (2023) represent a different editing paradigm not directly compatible with the current experimental setup, and adapting our framework to protect against them is a promising direction for future work.

Table 5: *Editing Models Used*

| Method | Editing Models Used |
|---|---|
| **DiffVax (Ours)** | SD Inpainting, IP2P, MagicBrush |
| **DiffusionGuard Choi et al. (2025) (ICLR 2025)** | SD Inpainting, IP2P |
| **PhotoGuard Salman et al. (2023) (ICML 2023)** | SD Inpainting, SDEdit |
| **SDS Xue et al. (2024) (ICLR 2024)** | SDEdit, SD Inpainting, Textual inversion |
| **Mist Liang & Wu (2023) (ICML 2023)** | Textual inversion, Dreambooth |
| **AdvDM Liang et al. (2023) (ICML 2023)** | Textual inversion, SDEdit |

### A.3 ADDITIONAL ROBUSTNESS EVALUATIONS AND STUDIES

#### A.3.1 ROBUSTNESS TO DIFFERENT SAMPLING STEPS AND SAMPLER SETTINGS

To evaluate the robustness of `DiffVax` against attackers who may vary their inference-time settings, we conduct experiments with different sampling steps and diffusion samplers. The results, presented in Table 6 and Table 7, demonstrate that `DiffVax` consistently and effectively disrupts malicious edits across a range of configurations.

Table 6 shows that `DiffVax` maintains superior performance across various sampling step counts (10, 20, 30, and 40). In nearly all scenarios, it achieves the best (lowest) scores for PSNR, FSIM, and CLIP-T, indicating its protection is not compromised when an attacker uses fewer or more steps for generation. Similarly, Table 7 illustrates that `DiffVax` outperforms baselines when different samplers (PNDMScheduler Liu et al., EulerDiscreteScheduler, LMSDiscreteScheduler Karras et al. (2022)) are used. This confirms that our learned immunization is not overfitted to a specific generation algorithm and remains effective in diverse, real-world attack scenarios.

Table 6: *Sampling Step Comparison*

| Sampling Step | Model | SSIM ↓ | PSNR ↓ | FSIM ↓ | CLIP-T ↓ |
|---|---|---|---|---|---|
| | PG-D | 0.637 | 16.79 | 0.391 | 26.54 |
| 10 | DiffusionGuard | 0.651 | 16.65 | 0.409 | 23.84 |
| | DiffVax | **0.627** | **16.37** | **0.366** | **22.96** |
| | PG-D | **0.564** | 15.56 | 0.379 | 28.89 |
| 20 | DiffusionGuard | 0.591 | 15.28 | 0.393 | 26.04 |
| | DiffVax | **0.564** | **14.96** | **0.360** | **24.42** |
| | PG-D | **0.523** | 14.92 | 0.379 | 29.27 |
| 30 | DiffusionGuard | 0.556 | 14.71 | 0.386 | 27.10 |
| | DiffVax | 0.526 | **14.32** | **0.362** | **24.17** |
| | PG-D | **0.507** | 14.42 | 0.377 | 29.68 |
| 40 | DiffusionGuard | 0.539 | 14.16 | 0.386 | 27.84 |
| | DiffVax | 0.506 | **13.78** | **0.356** | **24.06** |

Table 7: *Sampler Comparison*

| Sampler | Model | SSIM ↓ | PSNR ↓ | FSIM ↓ | CLIP-T ↓ |
|---|---|---|---|---|---|
| | PG-D | 0.480 | 14.31 | 0.404 | 26.88 |
| PNDMScheduler | DiffusionGuard | 0.501 | 14.52 | 0.404 | 26.97 |
| | DiffVax | **0.440** | **13.41** | **0.372** | **21.67** |
| | PG-D | 0.504 | 14.93 | 0.399 | 28.08 |
| EulerDiscreteScheduler | DiffusionGuard | 0.530 | 14.93 | 0.406 | 27.28 |
| | DiffVax | **0.466** | **13.91** | **0.361** | **22.00** |
| | PG-D | 0.487 | 14.36 | 0.403 | 27.82 |
| LMSDiscreteScheduler | DiffusionGuard | 0.509 | 14.47 | 0.405 | 27.23 |
| | DiffVax | **0.449** | **13.43** | **0.367** | **21.70** |

### A.3.2 IMMUNIZATION NOISE COMPARISON UNDER PERTURBATION BUDGET

We have run experiments comparing DiffVax's average learned perturbation against baselines with fixed 16/255, 32/255, 64/255 budgets, and we performed evaluation based on the mean magnitude ($L_1$) of the immunization noise (perturbation).

The results clearly show that DiffVax achieves superior edit disruption with a much smaller mean magnitude ($L_1$) perturbation than baselines given a larger budget, highlighting that its strength lies in the strategic placement of noise, not simply its magnitude. This supports our claim that DiffVax learns a more efficient and targeted noise distribution rather than applying uniform, high-energy noise.

Unlike methods that enforce a rigid and uniform $L_p$ budget, DiffVax implicitly learns the perturbation's properties via the trade-off in our loss function, $\mathcal{L} = \alpha \cdot \mathcal{L}_{\text{noise}} + \mathcal{L}_{\text{edit}}$. This allows the model to strategically allocate its "budget," applying stronger noise only where most effective and least visible.

Table 8: *Comparison Across Immunization Strengths ($\epsilon$)*

| $\epsilon$ | Method | SSIM ↓ | PSNR ↓ | FSIM ↓ | CLIP-T ↓ | SSIM (Noise) ↑ | Mean Magnitude (L1) of Immunization Noise ↓ |
|---|---|---|---|---|---|---|---|
| 64/255 | PG-D | **0.492** | 14.13 | 0.355 | 27.85 | 0.947 | 0.007 |
| | DiffusionGuard | 0.507 | 13.98 | 0.360 | 24.83 | 0.900 | 0.012 |
| 32/255 | PG-D | 0.502 | 14.23 | 0.360 | 29.18 | 0.950 | 0.006 |
| | DiffusionGuard | 0.526 | 14.30 | 0.373 | 26.13 | 0.927 | 0.009 |
| 16/255 | PG-D | 0.528 | 14.60 | 0.387 | 30.27 | 0.978 | 0.003 |
| | DiffusionGuard | 0.546 | 14.46 | 0.388 | 26.36 | 0.965 | 0.005 |
| – | DiffVax | 0.496 | **13.85** | **0.352** | **22.96** | **0.989** | **0.001** |

### A.3.3    ROBUSTNESS TO COUNTERATTACKS

JPEG compression and denoising techniques are typically designed to remove high-frequency components from images. Since our immunizer model introduces primarily low-frequency perturbations—due to the design of our noise loss—it becomes inherently more robust against such counterattacks.

Table 9 reports results under various JPEG compression ratios and when using IMPRESS (Cao et al., 2023), a model specifically developed for adversarial purification and denoising. Across all configurations, `DiffVax` consistently outperforms PhotoGuard-D and DiffusionGuard in terms of SSIM, SSIM (Noise), and CLIP-T metrics. These results suggest that `DiffVax` maintains its protective efficacy even when subjected to aggressive counterattack scenarios.

Figure 11 presents qualitative results of two counterattack strategies: (a) applying a denoiser and (b) applying JPEG compression. The edited image, along with its attacked counterpart, is shown for both PhotoGuard-D and `DiffVax`. While the visual changes for PhotoGuard-D are significant—indicating its vulnerability to counterattacks—`DiffVax` retains its robustness, preventing successful malicious edits.

To further explore robustness, Figure 19 presents additional qualitative comparisons under varying JPEG compression ratios (from 0.85 to 0.55) and under the IMPRESS purification attack. Even at high compression levels, `DiffVax` continues to disrupt the edits, showcasing its superior generalization and resistance to counter-editing.

Table 9: ***Additional counterattack experiments.*** The SSIM, SSIM (Noise), and CLIP-T metrics are reported for JPEG compression with ratios of 0.85, 0.65, and 0.55, as well as for the adversarial purification model IMPRESS. The metrics demonstrate that `DiffVax` consistently outperforms PhotoGuard-D (PG) and DiffusionGuard (DG), even when counterattacks are applied to all methods.

| Metric | DiffVax (JPEG .85) | DG (JPEG .85) | PG (JPEG .85) | DiffVax (JPEG .65) | DG (JPEG .65) | PG (JPEG .65) | DiffVax (JPEG .55) | DG (JPEG .55) | PG (JPEG .55) | DiffVax (IMPRESS) | DG (IMPRESS) | PG (IMPRESS) |
|---|---|---|---|---|---|---|---|---|---|---|---|---|
| SSIM $\downarrow$ | **0.517** | 0.646 | 0.640 | **0.530** | 0.696 | 0.692 | **0.534** | 0.706 | 0.693 | **0.489** | 0.605 | 0.578 |
| SSIM (Noise) $\uparrow$ | **0.968** | 0.955 | 0.961 | **0.951** | 0.946 | 0.950 | **0.944** | 0.940 | 0.944 | **0.644** | 0.636 | 0.640 |
| CLIP-T $\downarrow$ | **25.76** | 30.83 | 32.00 | **26.83** | 31.80 | 32.15 | **27.67** | 31.93 | 32.20 | **24.67** | 30.71 | 31.35 |

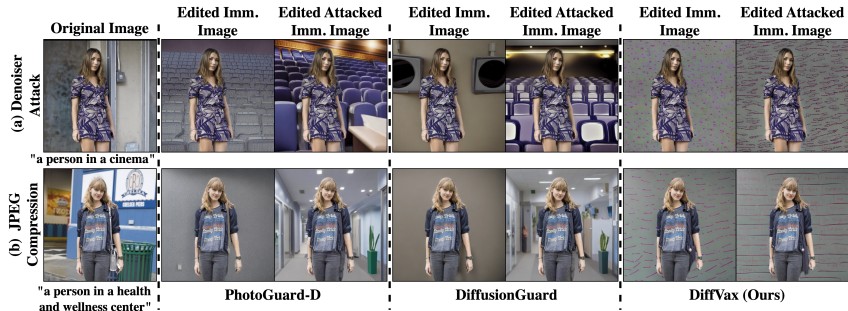

Figure 11: *Qualitative results of counter-attacks on immunization methods.* The first row shows results when an off-the-shelf denoiser is applied to the immunized image, while the second row displays results under JPEG compression. Columns 2–3 correspond to PhotoGuard-D, while columns 4–5 show results for `DiffVax`. PhotoGuard-D is visibly more susceptible to counterattacks, whereas `DiffVax` maintains strong protection.

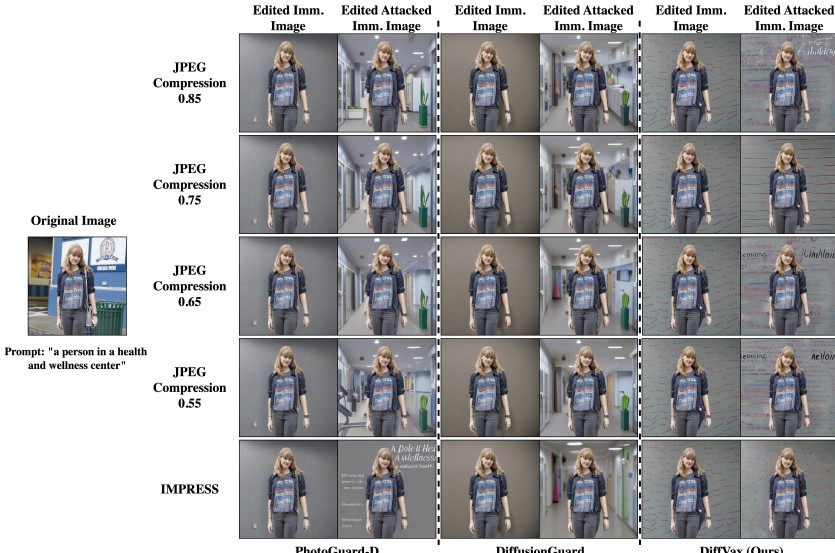

Figure 12: *Additional qualitative results of counter-attacks on immunization methods.* Each row corresponds to a different JPEG compression ratio or the IMPRESS model. `DiffVax` shows robust behavior across all levels, continuing to suppress harmful edits even under heavy degradation or purification.

### A.3.4 ROBUSTNESS TO NON-HUMAN SUBJECTS

To evaluate the generalizability of `DiffVax` beyond human-centric content, we conduct experiments on non-human subjects, such as animals and other inanimate objects. As illustrated in Figure 13, `DiffVax` effectively immunizes these non-person regions, preventing malicious edits while preserving the visual fidelity of the original image. These results further demonstrate the versatility and zero-shot capabilities of `DiffVax` across diverse object domains.

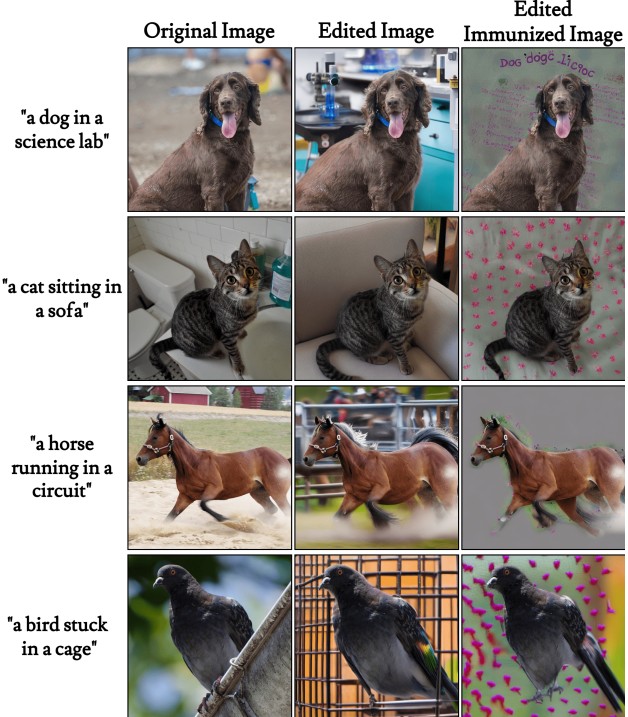

Figure 13: ***Qualitative results for non-human objects edited using `DiffVax`.*** These examples show that `DiffVax` extends effectively to domains beyond human subjects, maintaining its edit-resistance and imperceptibility.

### A.3.5 USER STUDY

To assess the human-perceived quality and effectiveness of each immunization method, we conducted a user study with 67 participants recruited via Prolific. Participants were asked to rank edited images based on how unrealistic or misaligned they appeared.

Each participant was shown a set of five edited images derived from the same input image and text prompt (see Figure 14). These five outputs corresponded to different immunization strategies: Random Noise, PhotoGuard-E, PhotoGuard-D, DiffVax, and an unprotected baseline. For each prompt-image pair, participants were instructed to rank the edits from **least aligned** to **most aligned** with the editing prompt. A lower ranking indicates better disruption of the intended edit (i.e., more effective immunization), as participants found the result less realistic or aligned with the prompt.

We randomly shuffled the order of methods in each trial to avoid position bias. In total, the study included 20 image-prompt pairs covering both seen and unseen examples, ensuring a fair and comprehensive evaluation.

Table 10: *User Study Rankings.* Lower values indicate better perceived editing failure prevention, imperceptibility, and alignment with the original content.

| Immunization Method | Average Ranking ↓ |
|---|---|
| Random Noise | 3.74 |
| PhotoGuard-E | 3.33 |
| PhotoGuard-D | *2.63* |
| DiffVax (Ours) | **1.64** |

As shown in Table 10, DiffVax significantly outperforms prior methods, receiving the best average ranking of **1.64**. This demonstrates the effectiveness of our method in fooling editing models in a way that is perceptually convincing to human observers. The next-best method, PhotoGuard-D, trails behind with a score of 2.63, while other methods rank even lower.

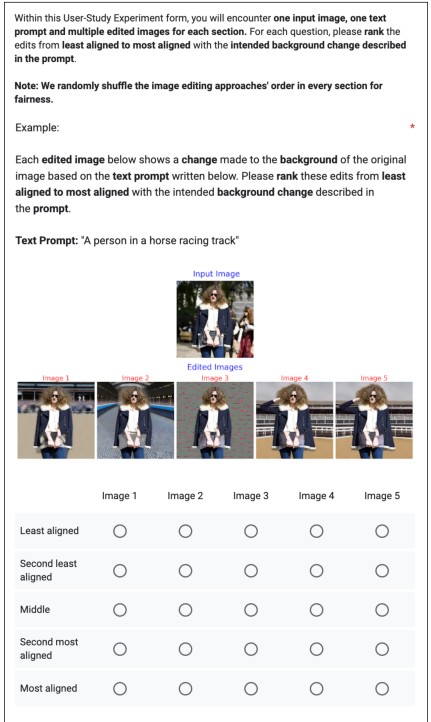

Figure 14: *Instructions provided to user study participants.* Users were asked to rank edited images from least to most aligned with the text prompt. Lower alignment suggests more successful immunization.

### A.3.6 VIDEO EVALUATION

To our knowledge, this is the first immunization-based video evaluation using a diffusion model for editing. We construct a video benchmark consisting of 4 human activity videos, each containing 64 frames and paired with 4 unique prompts. Since no prior method directly supports training-free video immunization using inpainting-based diffusion models, we adopt a naive per-frame editing pipeline to extend our approach to video. Despite not incorporating any explicit temporal modeling, our method yields strong results.

As reported in Table 11, `DiffVax` outperforms all baselines across multiple metrics, including PSNR, SSIM (Noise), CLIP-T, and runtime. Notably, it achieves a dramatic reduction in runtime—processing the full dataset in just **0.739 seconds**—compared to PhotoGuard-D's 64-hour runtime. These results emphasize the efficiency and practicality of our approach in real-time or large-scale settings.

Importantly, we make no architectural or training modifications for video data. The strong results achieved without temporal modeling suggest that our method generalizes well across sequential data, capturing consistent patterns in human identity, pose, and structure across frames. This robustness is further demonstrated in Fig. 1 and Fig. 4 (c), where the model effectively adapts to changes in body motion and facial expressions.

Our work targets general-purpose editing protection and is evaluated on diverse, open-domain video data. The effectiveness of our approach under such settings demonstrates its promise as a scalable and general immunization strategy for future video editing systems.

Table 11: ***Results on video editing.*** We report the average PSNR, SSIM, FSIM, SSIM (Noise), CLIP-T, and total runtime for Random Noise, PhotoGuard-D, DiffusionGuard, and `DiffVax` on a video dataset consisting of 4 videos, each with 4 prompts and 64 frames. Best results per column are **bolded**.

| Method | SSIM ↓ | PSNR ↓ | FSIM ↓ | SSIM (Noise) ↑ | CLIP-T ↓ | Runtime ↓ |
|---|---|---|---|---|---|---|
| Random Noise | 0.774 | 21.09 | 0.547 | 0.786 | 29.62 | N/A |
| PhotoGuard-D | 0.738 | 17.31 | 0.448 | 0.965 | 26.52 | 64 hours |
| DiffusionGuard | 0.750 | 17.43 | 0.478 | 0.922 | 25.41 | 10 hours |
| DiffVax | **0.681** | **16.78** | **0.374** | **0.974** | **22.51** | **0.739 seconds** |

## A.4 Discussion on Generalization

While a universal immunizer that works zero-shot across all editing model architectures is a challenging open problem, `DiffVax` demonstrates superior generalization compared to existing optimization-based methods across three distinct dimensions: generalization to unseen models, to unseen content, and to unseen masks. This section details these advantages.

### A.4.1 Generalization to Unseen Models

Existing immunization methods, including optimization-based approaches like PhotoGuard, are model-specific. While developing a universally transferable immunizer is not the primary focus of this work, `DiffVax` demonstrates significantly better generalization to unseen models than prior methods. We conducted an experiment where immunization noise was generated using a model trained on Stable Diffusion (SD) v1.5 and then tested on an unseen SD v2 model. As shown qualitatively in Figure 15, `DiffVax` successfully transfers its protective effect, whereas PhotoGuard's perturbations fail completely, leaving the image vulnerable.

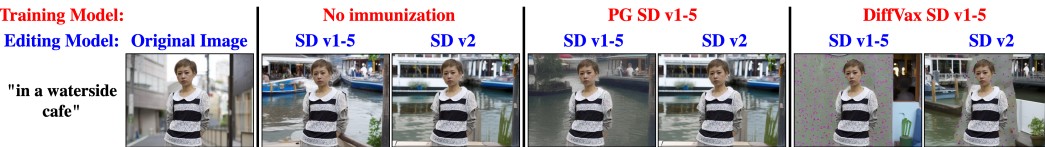

Figure 15: ***Transferability of perturbations across editing models.*** Red labels indicate the immunization training model, and blue labels denote the editing model. The results show how well each immunized image resists edits across different model configurations. When trained on Stable Diffusion (SD) v1.5, `DiffVax` successfully prevents edits even when tested on SD v2. In contrast, PhotoGuard's perturbations trained on SD v1.5 do not generalize to SD v2. These results illustrate the superior cross-model generalizability of `DiffVax`.

Table 12 provides quantitative results for this black-box transfer task, confirming that `DiffVax` achieves the best performance across all metrics. This provides direct evidence that our learned immunization strategy is more robust and generalizable across model versions than optimization-based approaches.

Table 12: Quantitative results for transferring immunization from SD v1.5 to an unseen SD v2.0 model. Lower values are better for all metrics, indicating more effective edit disruption. `DiffVax` outperforms all baselines.

| SD 2.0 | SSIM ↓ | PSNR ↓ | FSIM ↓ | CLIP-T ↓ |
|---|---|---|---|---|
| PG-D | 0.566 | 15.17 | 0.417 | 32.00 |
| DiffusionGuard | 0.609 | 15.26 | 0.454 | 31.73 |
| **DiffVax** | **0.540** | **14.02** | **0.384** | **27.72** |

### A.4.2 Generalization to Unseen Content

Optimization-based methods inherently handle unseen images by running a costly, per-image optimization process. A key scientific question this paper addresses is whether it is possible to learn a single feed-forward model that can directly generate effective perturbations without optimization. The success of our approach implies that the set of effective perturbations across all possible images possesses sufficient structure and regularity to be learnable. Our experiments demonstrate that `DiffVax` successfully generalizes to **unseen images, unseen prompts, and even unseen videos** with a single forward pass, as demonstrated in Fig. 1 and Fig. 4 (b) and (c) and Table 11. This establishes the learnability of the perturbation set for the first time and enables protection at a scale and speed previously unattainable.

### A.4.3 GENERALIZATION TO UNSEEN MASKS DURING TEST TIME

Most existing state-of-the-art (SOTA) methods assume that the same mask is used during both the immunization (training) and editing (testing) phases. While this assumption aligns with standardized deepfake pipelines—where masks are often fixed to cover specific regions such as the head or full body—it limits the robustness of these methods to real-world scenarios involving unpredictable or mismatched editing masks.

To evaluate this limitation, we conduct an experiment where the editing mask during test time differs from the mask used during immunization. As shown in Figure 17, when the test-time mask diverges from the training mask, existing methods such as PhotoGuard (PG) and DiffusionGuard fail to maintain their edit-disrupting behavior. In contrast, `DiffVax` remains effective, successfully disrupting the malicious edits even when significant changes are made to the mask size or region. This robustness can be attributed to our model's design, which does not overfit to the spatial shape or scale of the mask used during training. Instead, it learns to encode more generalizable perturbations that degrade editing attempts across a range of editing contexts. These findings suggest that `DiffVax` offers better real-world applicability where attackers may alter masks to evade immunization. We further demonstrate this in Figure 16, which illustrates that `DiffVax` maintains its protective capabilities even when the test-time mask only partially covers the subject (e.g., a vertical half-mask), preventing the edit despite the minimal mask overlap.

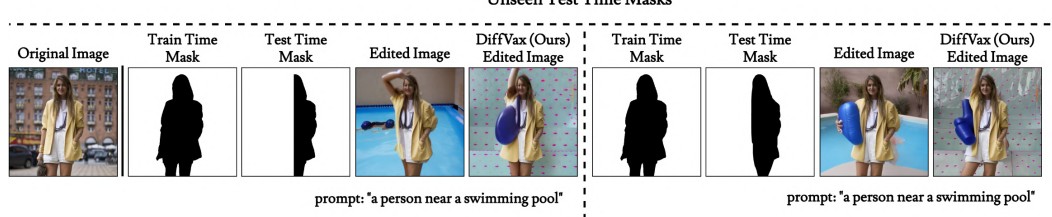

Figure 16: *Additional comparison of partially covering masks.* We evaluate robustness in scenarios where the test-time mask only covers a portion of the subject (e.g., a vertical half-split) compared to the full mask used during immunization. As shown, `DiffVax` successfully maintains protection and disrupts the edit despite the partial mask coverage.

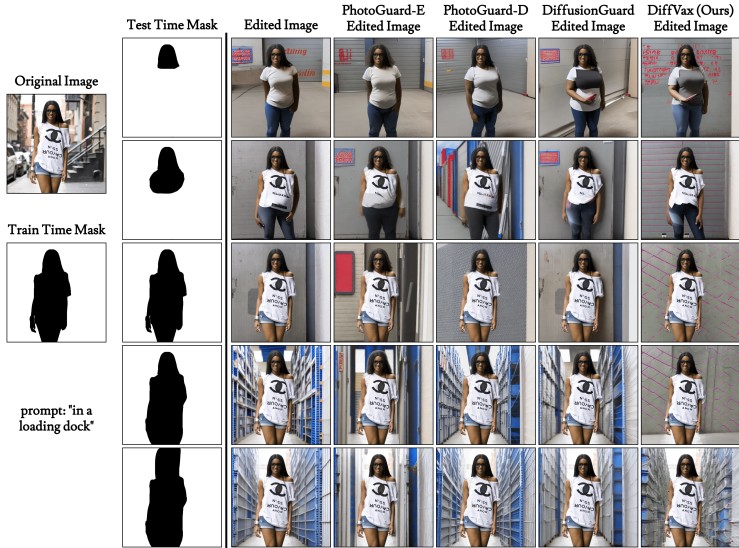

Figure 17: *Comparison of edited immunized images with different immunization and editing masks.* PhotoGuard uses the same mask for both training and testing, making it highly sensitive to changes in the editing mask. `DiffVax`, by contrast, is trained with a fixed immunization mask but remains robust even when the test-time editing mask significantly deviates. The results show consistent disruption of edits by `DiffVax` despite large mask variability.

### A.5 Imperceptibility Discussion

To evaluate the imperceptibility of the perturbations introduced by `DiffVax`, we present qualitative comparisons against PhotoGuard in Figure 18. Our method generates noise that is concentrated in the low-frequency components of the image, making it visually more subtle and less disruptive. In contrast, PhotoGuard introduces high-frequency noise that appears scattered across broader regions.

This low-frequency characteristic of `DiffVax` offers two key advantages. First, it enhances the perceptual quality of the immunized images by producing smoother perturbations that minimally interfere with semantic content. Second, it contributes to robustness against counterattacks such as JPEG compression or denoising—these techniques are typically designed to suppress high-frequency information, which is assumed to correspond to noise. Since `DiffVax` avoids relying on high-frequency artifacts, its perturbations are more likely to survive such transformations, preserving the protective effect.

We further examine the role of the loss norm in shaping the visual quality of the immunization. As shown in Figure 19, using $L_2$ or $L_\infty$ norms leads to less perceptible perturbations than the default $L_1$ formulation. However, this comes at the expense of reduced edit resistance, underscoring a critical trade-off between imperceptibility and robustness.

Future work will explore more principled approaches to navigating this trade-off, such as incorporating perceptual similarity metrics or frequency-domain regularization directly into the optimization objective.

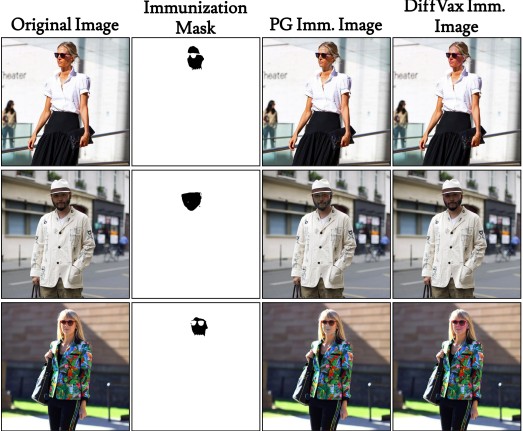

Figure 18: ***Comparison of immunization noise.*** Visual comparison of immunized images generated by PhotoGuard and `DiffVax` using a face mask. PhotoGuard produces scattered and higher-frequency noise, while `DiffVax` generates smoother, low-frequency perturbations.

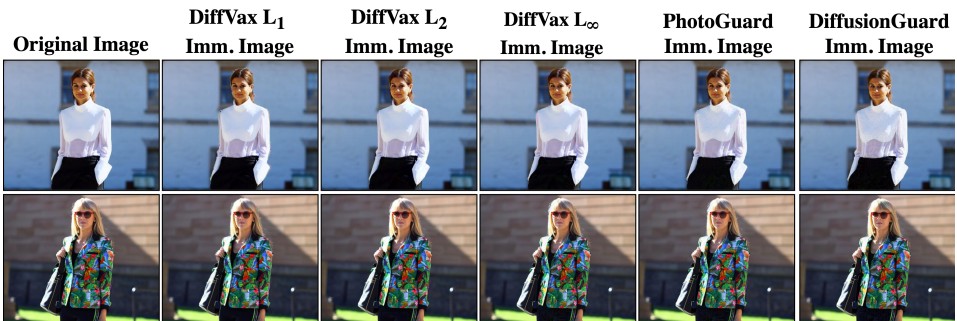

Figure 19: ***Additional comparison of immunization noise under different norms.*** This figure compares immunized images generated using different norm constraints: $L_1$, $L_2$, and $L_\infty$, as well as results from PhotoGuard and DiffusionGuard.

### A.6 PROMPT-AGNOSTIC IMMUNIZATION EXPERIMENT

We conduct additional experiments to demonstrate that the noise produced by our `DiffVax` (and consequently the immunized images) is prompt-agnostic. To achieve this, we train `DiffVax` three times, using a different image for each training setup. In each experiment, we use a single image with 100 seen prompts for training and evaluate it on 75 seen prompts and 75 unseen prompts (not included in the training set). The results are then averaged across all images for each prompt. As shown in Fig. 20, the quantitative results for seen and unseen metrics are highly similar, and the low variances further confirm that the noise generalizes effectively across diverse prompt conditions.

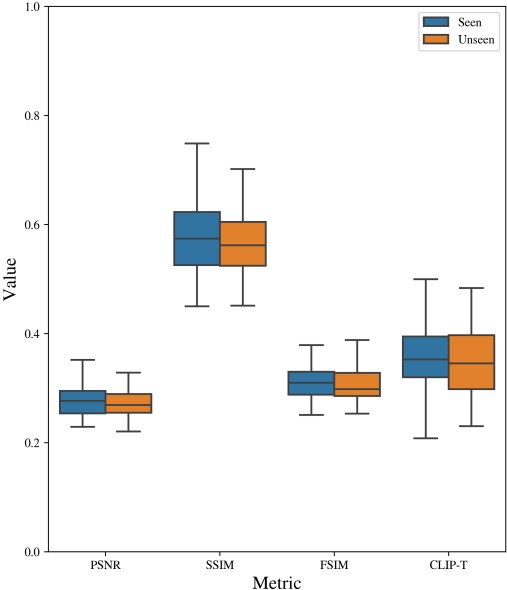

Figure 20: ***Experiment results for prompt-agnostic noise.*** We present our performance metrics between prompts for 75 prompts seen in training (blue color) and 75 prompts unseen in training (orange color). PSNR and CLIP-T values are divided by 50 for visualization purposes. We can see that the two distributions are almost identical, suggesting that our method performs similarly across all prompts, suggesting the prompt-agnostic nature of our `DiffVax`.

### A.7 LOSS WEIGHT SELECTION

The hyperparameter $\alpha$ in `DiffVax`'s loss function controls the balance between imperceptibility and edit disruption. It is defined in the overall loss as $\mathcal{L} = \alpha \cdot \mathcal{L}_{\text{noise}} + \mathcal{L}_{\text{edit}}$, where a larger $\alpha$ emphasizes minimizing visible noise, potentially at the cost of reduced editing resistance, and a smaller $\alpha$ enhances robustness to edits but may introduce more perceptible perturbations.

To determine an optimal value for $\alpha$, we conduct an ablation study on a subset of 100 images, evaluating three values: $\alpha = 2$, 4, and 6. The results are summarized in Table 13. We observe that while increasing $\alpha$ improves imperceptibility—as indicated by slightly higher SSIM (Noise) and PSNR scores—the edit disruption becomes weaker, reflected in a deterioration of the SSIM and PSNR metrics.

We select $\alpha = 4$ as the optimal configuration. It provides a strong balance between imperceptibility and disruption: the gain in SSIM (Noise) from $\alpha = 4$ to $\alpha = 6$ is marginal, while the drop in editing robustness is more pronounced. Furthermore, qualitative inspection confirms that the perturbations at $\alpha = 4$ are already imperceptible, making further increase in $\alpha$ unnecessary.

Table 13: *Ablation study on the loss weight $\alpha$ in $\mathcal{L} = \alpha \cdot \mathcal{L}_{noise} + \mathcal{L}_{edit}$.* Metrics demonstrate the trade-off between imperceptibility and edit disruption. Best values for SSIM (Noise) are bolded, while lower SSIM and PSNR indicate stronger editing disruption.

| Configuration | SSIM ↓ | PSNR ↓ | SSIM (Noise) ↑ |
|---|---|---|---|
| `DiffVax` w/ $\alpha = 2$ | **0.536** | **14.47** | 0.987 |
| `DiffVax` w/ $\alpha = 4$ | 0.588 | 15.38 | **0.993** |
| `DiffVax` w/ $\alpha = 6$ | 0.625 | 16.23 | **0.996** |

## A.8 LIMITATIONS

Despite its strong performance, `DiffVax` exhibits certain limitations, as illustrated in Figure 21. First, the model faces challenges in effectively immunizing scenes with multiple small objects (Panel A). In such cases, the protective noise may be too dispersed to disrupt the semantic guidance for every individual object, allowing some edits (e.g., turning birds into parrots) to succeed. Second, while generally robust to mask variations, extreme discrepancies between the immunization and editing masks can compromise protection. For instance, if an image is immunized with a full-body mask but the attacker employs a significantly smaller, localized mask (Panel B), the edit may partially bypass the immunization. Finally, although we optimize for imperceptibility, occasional perceptible artifacts may appear (Panel C), particularly in smooth or uniform regions where the adversarial perturbations are harder to conceal.

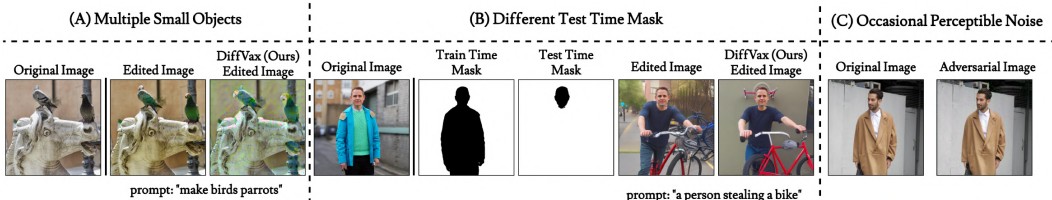

Figure 21: *Limitations.* We highlight three failure cases: (A) `DiffVax` struggles to protect multiple small objects simultaneously; (B) protection may fail when the test-time editing mask (e.g., a small face region) drastically differs from the training-time immunization mask (e.g., full body); and (C) in rare instances, the immunization noise becomes perceptible to the human eye.

## A.9 FUTURE DIRECTIONS

While DiffVax demonstrates strong generalization within the latent diffusion paradigm, a critical avenue for future research is achieving universal cross-architecture immunization. Current perturbations are optimized for the gradient flow of U-Net-based LDMs, which may not fully transfer to emerging non-LDM architectures, such as flow-matching models or pixel-space diffusion transformers. We envision extending our scalable feed-forward framework to incorporate *Ensemble Adversarial Training*, where the immunizer is trained simultaneously against a diverse set of backbones. This would encourage the model to learn universally disruptive features that transcend specific architectural biases. Additionally, expanding the training distribution to include non-photorealistic domains, such as anime and digital art, will be essential for broader real-world applicability.

Furthermore, while our method effectively immunizes video via per-frame processing, future iterations could explicitly model temporal dynamics. Integrating temporal consistency objectives directly into the training loop would allow the immunizer to exploit motion priors, potentially creating "video-native" perturbations that are even more robust to temporal filtering and video compression. Finally, addressing the physical limitations of extremely small editing masks remains an open challenge; exploring frequency-domain regularization or perceptual loss functions that adaptively balance noise visibility with protective strength in constrained regions could offer a pathway to more granular control.

## A.10 REPRODUCIBILITY STATEMENT

The source code of the project is provided in the supplementary. Project can be reproduced by following the provided guidelines and source code. All experiments can be replicated using the instructions and datasets referenced in this paper.

## A.11 ETHICS STATEMENT

This work does not raise any foreseeable ethical concerns. The experiments were conducted solely on publicly available datasets.

## A.12 LLM USAGE STATEMENT

Large language models (LLMs) were used exclusively for assistance in grammar correction, formatting, and improving the clarity of writing. They were not employed for generating research ideas, designing experiments, or creating results.