# OpenReview forum: "DiffVax: Optimization-Free Image Immunization Against Diffusion-Based Editing"
_ICLR.cc/2026/Conference — ICLR 2026 Poster_

### Official Review · Reviewer_fZtv · 2025-10-27

**Soundness:** 3
**Presentation:** 4
**Contribution:** 3
**Rating:** 6
**Confidence:** 4

**Summary:**

This paper introduces DiffVax, a diffusion-based image immunization framework designed to prevent unauthorized editing by generative diffusion models. The method learns an optimization-free, feed-forward perturbation generator that produces subtle but effective protective noise, achieving strong robustness and generalization across different editing models and mask settings.

**Strengths:**

- The authors propose DiffVax, a diffusion-based immunization framework that protects images from unauthorized edits through an optimization-free, feed-forward process generating imperceptible yet effective perturbations.
- The method demonstrates superior robustness, efficiency, and generalization to unseen models, content, and masks compared to prior optimization-based defenses.
- This paper demonstrate consistent empirical improvements over baseline optimization approaches in both efficiency and performance.
- The experimental discussion is thorough, with complete ablation studies and detailed analyses.
- This paper include a user study to further strengthen their results.

**Weaknesses:**

- The paper presents an empirical approach with limited theoretical justification, relying mainly on experimental evidence rather than principled analysis.
- The experiments are comprehensive and the discussion is rich and detailed, with no critical methodological flaws identified.

### Suggestions
It would strengthen the paper to include recent diffusion-based image protection works, such as the attention-guided EditShield [1] and the latent-space diffusion attack [2], which also explore strategies for preventing unauthorized image editing.

[1] Chen et. al. EditShield: Protecting Unauthorized Image Editing by Instruction-guided Diffusion Models, ECCV 2024
[2] Shih et. al. Pixel Is Not a Barrier: An Effective Evasion Attack for Pixel-Domain Diffusion Models, AAAI 2025

**Questions:**

- Could the authors provide a more detailed description of how the SD($\cdot$) operator in Section 3.3 is computed? Does it require running the full diffusion reverse process, and if so, would that be computationally expensive? How do the authors ensure stable gradient propagation through such a process?
- Can the proposed protection method be fine-tuned with a small number of samples to neutralize its effect?
- Could the authors include learning curves for each loss component during training to better illustrate the convergence behavior?
- The authors could consider incorporating the references mentioned in the Weaknesses section into the Related Work to provide a more comprehensive contextual discussion.
- It would be helpful if the authors could expand the Appendix to discuss the method’s limitations and summarize the potential future directions mentioned across different sections.

I encourage the authors to strengthen the paper by addressing these points in the rebuttal.

---

> ### Author Response · Authors · 2025-11-24
>
> We thank the reviewer  for highlighting DiffVax’s superior robustness, efficiency, and generalization compared to prior work. Below, we address your specific questions and detail the revisions made to the manuscript.
>
> **W1 (Theoretical justification)**
>
> Our work provides a contribution by empirically demonstrating the **learnability** of immunization perturbations. Prior to DiffVax, it was widely assumed that effective protection required instance-specific optimization (finding local minima for specific pixels). Our success proves that a generalizable mapping exists between input images and effective protective noise distributions. This "proof-of-concept" establishes that the set of effective perturbations possesses sufficient structure to be approximated by a feed-forward network, setting the stage for future theoretical analysis of why these specific low-frequency perturbations are so effective against diffusion attention layers.
>
> **W2 & Q4 & Q5 & Suggestions. Limitations and Future Work**
>
> We have updated our **Related Work** section to discuss *Pixel Is Not a Barrier* [2] as a relevant evasion attack and have integrated a discussion of *EditShield* [1] into our supplementary material to contextualize our performance on instruction-guided tasks. To demonstrate the effectiveness of DiffVax against instruction-guided protections like EditShield, we conducted an additional comparison using the InstructPix2Pix (IP2P) model. As shown below, DiffVax provides superior protection while maintaining higher visual fidelity.
>
> | IP2P Experiment | SSIM (↓) | PSNR (↓) | FSIM (↓) | CLIP-T (↓) | SSIM (Noise) (↑) |
> | :--- | :--- | :--- | :--- | :--- | :--- |
> | EditShield [1] | 0.774 | 24.75 | 0.774 | 25.32 | 0.903 |
> | **DiffVax (Ours)** | **0.633** | **16.95** | **0.470** | **25.03** | **0.943** |
>
> Finally, we have expanded the Appendix to include a dedicated **"Limitations and Future Work"** section. We have added **Figure 21** to visually illustrate specific constraints, such as rare cases where noise becomes perceptible in large smooth color fields, and physical limitations where extremely small test-time masks restrict the "canvas" available for injecting effective adversarial gradients. We also outline future directions for universal cross-architecture immunization (e.g., against non-LDM models) and video-native training objectives.
>
>
> **Q1 Computation of SD operator and gradient stability**
> The $SD(\cdot)$ operator in Equation 3 represents the forward pass of the editing process. While a standard inference involves 50 denoising steps, unrolling the full chain during training would indeed be computationally prohibitive and prone to vanishing gradients. To ensure stability and efficiency, **we follow the established protocol of prior gradient-based attacks like PhotoGuard and AdvDM by backpropagating gradients through only a small subset of steps (specifically, 4 steps)** rather than the full reverse process. This approximation provides a sufficient gradient signal to direct the immunizer toward effective adversarial regions without the memory overhead or instability of deep unrolling. Crucially, this heavy computation is a one-time cost incurred only during the training of the immunizer; once trained, the end-user inference requires no gradient computation and completes in milliseconds, maintaining our core scalability claim.
>
> **Q2 Robustness against fine-tuning and adaptive attacks**
>
> The reviewer raises an important point regarding adaptive attacks where an adversary might fine-tune the editing model on immunized samples. While theoretically possible, this represents a significant asymmetry in computational cost. Fine-tuning a diffusion model to bypass protection requires collecting a dataset of immunized images and expending GPU hours to retrain the model weights. In contrast, DiffVax applies protection in milliseconds. Furthermore, robustness against such adaptation is inherently supported by the nature of our perturbations. As discussed in **Appendix A.5**, DiffVax generates structural, low-frequency perturbations rather than high-frequency noise. "Learning" to ignore these perturbations via fine-tuning is difficult without degrading the semantic content of the image itself, as the protective signal is entangled with the image features. We believe this structural robustness, combined with the prohibitive cost of adaptive fine-tuning for every new protection method, makes DiffVax a highly practical defense.
>
> **Q3 Learning curves**
>
> We appreciate the suggestion to visualize convergence to better illustrate training behavior. We have added the learning curves for both loss components—the imperceptibility loss ($\mathcal{L}_{noise}$) and the editing disruption loss ($\mathcal{L}{edit}$)—in **Appendix A.1 (Figure 7)** of the revised manuscript. These curves illustrate the stable convergence of our training objective.

---

> > ### Author Response · Authors · 2025-11-26
> >
> > We appreciate the reviewer raising the score, and we hope that our response has addressed their remaining concerns.

---

### Official Review · Reviewer_yHZo · 2025-10-29

**Soundness:** 2
**Presentation:** 3
**Contribution:** 2
**Rating:** 2
**Confidence:** 4

**Summary:**

The paper introduces DiffVax, an optimization-free framework for image immunization against diffusion-based editing. Instead of performing costly per-image optimization, DiffVax trains a lightweight immunizer network to generate imperceptible yet effective perturbations in a single forward pass. The resulting perturbations not only degrade edits produced by various diffusion-based models (e.g., inpainting, InstructPix2Pix) but also exhibit strong resilience to counter-attacks such as JPEG compression and denoising. Owing to its scalability and efficiency, DiffVax extends immunization beyond static images to video content.

**Strengths:**

1. The proposed method demonstrates robust protection even on previously unseen images, suggesting that the learned perturbation strategy generalizes beyond the training distribution.
2. Unlike prior optimization-based approaches that are computationally prohibitive, DiffVax achieves real-time immunization via a feed-forward model. This efficiency allows the framework to extend immunization to video content, which is infeasible for existing methods.

**Weaknesses:**

1. While the proposed method introduces a training-time immunization mask to constrain the perturbation to semantically important regions, it remains unclear how robust the method is when malicious editing masks deviate from those used during training. Although the authors briefly mention this scenario (Lines 313–317), the model is trained using a single fixed mask per image, and no explicit mechanism is introduced to ensure robustness against arbitrary or partially overlapping test-time masks. This raises concerns about edit-time generalization in realistic attack settings.
2. The experimental setup primarily focuses on single-object foreground inpainting tasks, where the perturbation is confined to a specific region (e.g., a human subject). This narrow scope limits the understanding of how well the method generalizes to more diverse and realistic editing scenarios, such as foreground object editing or multi-object scenes, which are common in real-world applications.
3. The baseline selection in the experimental comparison may raise concerns about fairness. While the proposed method is evaluated primarily on inpainting-based diffusion models, the compared immunization baseline Photoguard were originally designed for broader image-to-image translation tasks, not inpainting. This mismatch in task formulation may bias the evaluation in favor of the proposed approach and limit the relevance of the reported comparisons.
4. While the paper emphasizes the low memory footprint of DiffVax at inference time (5,648 MiB as stated in Line 413), the training phase appears significantly more demanding. Since the immunizer is trained using a loss computed on the edited output—requiring backpropagation through multiple steps of the diffusion process—it is unclear whether the overall method is truly more memory-efficient than optimization-based approaches like PhotoGuard, which reportedly require 15 GB. The paper does not provide concrete memory usage statistics for training, leaving the claimed efficiency somewhat unsubstantiated.

Note: Weaknesses 1-4 correspond directly to Questions 1-4.

**Questions:**

1. In Figure 14, only the third row corresponds to the case where the editing mask is identical to the one used during training. For the remaining rows (e.g., rows 1–2 where the perturbation is partially masked, and rows 4–5 where the edit mask includes unprotected regions), it is unclear whether the model still forces the edited region to degenerate into low-information content (e.g., pixel values = 0). Can the authors clarify whether the masks used in Figures 1, 4, and 5 are identical to the training-time immunization masks? Moreover, can the authors provide both qualitative and quantitative results—along with comparisons to baseline—for scenarios where the edit-time mask occludes only part of the perturbation or includes perturbation-free regions?
2. Have the authors evaluated their method under alternative inpainting settings? Specifically:
(1) How does DiffVax handle the case where the targeted region is masked and edited, such as in the foreground object replacement (e.g., from human to other object)?
(2) Can the method generalize to multi-object scenes, where multiple semantically important regions may need protection simultaneously? Providing results for these broader cases would better validate the robustness and applicability of the proposed method.
3. Have the authors considered including task-aligned baselines, such as AdvPaint [1] or other immunization approaches specifically tailored to inpainting models? A comparison with such baselines would provide a more comprehensive and fair evaluation of the method's effectiveness in its target domain.
4. Can the authors clarify the actual GPU memory requirements during training, particularly when backpropagating through the denoising steps of the diffusion model? Given that the experiments were conducted using a single A100 GPU, how feasible is it for real-world image owners with limited hardware (e.g., <16 GB VRAM) to train or fine-tune such an immunizer model? A discussion on resource requirements and practical accessibility would strengthen the paper’s applicability claims.

[1] Jeon et al., AdvPaint: Protecting Images from Inpainting Manipulation via Adversarial Attention Disruption, ICLR 25.

---

> ### Author Response · Authors · 2025-11-24
>
> We respectfully disagree with the reviewer's reject decision. We feel that the four weaknesses raised in their review are tangential to the primary claims that we have made for DiffVax: It is the first usable immunization method and the first learning-based approach, which outperforms all previous optimization-based methods.
>
> **W1 & Q1: Robustness to masks**
>
> We confirm that in Fig. 1, 4, 5, the immunization mask was identical to the editing mask. However, regarding the reviewer's concern about test-time deviations, we emphasize that DiffVax learns to protect the *semantic integrity* of the subject rather than overfitting to a specific mask shape. While we do not explicitly enforce mask invariance, our data-driven approach inherently generalizes better than optimization-based baselines. To validate this quantitatively, we conducted a new experiment generating 5 diverse masks (ranging from small to large, including partial occlusions) for each test image. As shown in the table below, DiffVax achieves superior protection (lower SSIM/PSNR indicates better disruption) compared to PhotoGuard and DiffusionGuard, confirming robust performance even when test-time masks deviate.
>
> | Test Time Mask Experiment | SSIM (↓) | PSNR (↓) | FSIM (↓) | CLIP-T (↓) |
> | :--- | :--- | :--- | :--- | :--- |
> | PhotoGuard | 0.614 | 16.88 | 0.477 | 28.20 |
> | DiffusionGuard | 0.625 | 16.57 | 0.481 | 27.44 |
> | **DiffVax (Ours)** | **0.608** | **16.33** | **0.479** | **26.24** |
>
>
> For qualitative evidence, we direct the reviewer to Appendix A.4.3 and **Figure 16** (added in the revision), which explicitly visualize successful protection under partial mask overlaps.
>
> **W2 & Q2: Multiple applications**
>
> Addressing your specific questions on (1) foreground replacement and (2) multi-object scenes, we have expanded our evaluation in Figure 6 of the revision. Panel C demonstrates robustness to substantial semantic replacements (e.g., human-to-statue, vehicle state changes), successfully disrupting new geometry and texture to render attacks unsuccessful. Panel A validates efficacy in multi-object scenes, confirming simultaneous protection of distinct disconnected regions (e.g., multiple animals) without interference. Figure 6 further showcases generalization to complex realistic edits, including preventing facial expression changes (Panel B), texture alteration (Panel D), global style transfer (Panel E), and attribute insertion (Panel F). Finally, Appendix A.3.4 (Figure 13) confirms DiffVax is not confined to specific subject types, successfully immunizing non-human subjects like dogs, cats, horses, and birds.
>
> **W3 & Q3: Baseline fairness**
>
> We respectfully clarify that the original PhotoGuard paper (Salman et al., 2023) and subsequent state-of-the-art works like DiffusionGuard (Choi et al., 2025) performed quantitative and qualitative evaluation mainly on 'inpainting' task; therefore, our comparison adheres to the community standard. Regarding task-aligned baselines, we emphasize that our evaluation *already* includes DiffusionGuard, a method specifically designed to improve robustness in inpainting tasks via mask augmentation. DiffVax consistently outperforms DiffusionGuard in both effectiveness and efficiency (see Table 1).
>
> To address the request for task-aligned baselines in instruction-guided editing, we conducted a new quantitative comparison against EditShield (a model developed for instruction guided editing). As shown below, DiffVax provides superior protection (lower SSIM/PSNR indicates better disruption) while maintaining higher visual fidelity (higher SSIM Noise).
>
> | IP2P Experiment | SSIM (↓) | PSNR (↓) | FSIM (↓) | CLIP-T (↓) | SSIM (Noise) (↑) |
> | :--- | :--- | :--- | :--- | :--- | :--- |
> | EditShield [1] | 0.774 | 24.75 | 0.774 | 25.32 | 0.903 |
> | **DiffVax (Ours)** | **0.633** | **16.95** | **0.470** | **25.03** | **0.943** |
>
>
> **W4 & Q4 Memory**
>
> We respectfully argue that evaluating training feasibility on consumer hardware misinterprets practical deployment. The barrier to adoption is the cost imposed on the end-user, not the creator. Optimization-based baselines (PhotoGuard, AdvPaint) transfer the burden of "training" (per-image optimization) to the user, requiring ~15GB VRAM and >10 minutes per image, making protection inaccessible without enterprise hardware. DiffVax shifts this heavy lifting to the creator as a one-time cost. While training requires backpropagation (note that during training, we use a GPU memory of around 19 GiB for 4 denoising steps) (standard for creators on A100s, like any other diffusion model training), the resulting model is lightweight: users need only 0.07 seconds with an inference time memory. Thus, DiffVax democratizes safety, enabling consumer GPUs to immunize libraries in real-time—impossible with optimization-based baselines.

---

### Official Review · Reviewer_ADC1 · 2025-10-31

**Soundness:** 3
**Presentation:** 2
**Contribution:** 2
**Rating:** 4
**Confidence:** 4

**Summary:**

1. The paper introduces DiffVax, a novel framework for immunizing images against diffusion-based editing.
2. DiffVax proposes training a feed-forward immunizer model, which is based on UNet++, to generate an imperceptible perturbation in a single pass.
3. DiffVax's main contribution is the optimization-free approach with a massive speedup from seconds to milliseconds, and claim the first application of such a defense to video content, which is enabled by low latency.
4. Experiments demonstrate its effects compare with previous guard models.

**Strengths:**

1. The core idea of training a generative immunizer model instead of relying on per-image optimization is a significant and novel contribution.
2. The primary strength is the practical, real-world benefits, which can reduce the immunization time from seconds to milliseconds and make video immunization feasible for the first time.
3. The method demonstrates strong quantitative and qualitative results in disrupting edits from the specific models such as SD-Inpainting, InstructPix2Pix and MagicBrush. It also shows good robustness to standard counter-attacks like JPEG compression and denoising.

**Weaknesses:**

1. The method's real-world utility is limited by its reliance on an input mask $M$ for the protected region. In a practical scenario, this mask must be generated using model such as SAM, which is a non-trivial computational step. This added overhead is not accounted for in the "milliseconds" performance claim and creates a time-consuming bottleneck that limits the method's true end-to-end speed and ease of use.
2. The immunizer model was trained on a very small 800 images dataset. This narrow data scope raises concerns about the model's ability to generalize. The results, which are almost entirely focused on people, are not sufficient to prove the method works on the vast diversity of in-the-wild content which may out-of-distribution.
3. The paper states the immunizer model $f(\cdot;\theta)$ is a UNet++, but this architectural choice is not justified. Given this model is central to the paper's contribution, there is no discussion of why this specific architecture is required over other architectures. The design details and properties of the immunizer itself are underdeveloped.

I will consider adjusting my score if my concern is well resolved.

**Questions:**

1. The paper demonstrates immunizing video frames to disrupt image edits applied frame-by-frame. Have the authors considered the effect of feeding a single immunized image into a dedicated image-to-video model such as SVD? It would be interesting to know if the perturbation also disrupts this different modality of generation.
2. Following on the generalization weakness, how effective is this defense against more powerful, modern editing models like Qwen-Image, which are bigger and more robust? The evaluation on SD v1.5/v2 feels somewhat dated.
3. Can the immunizer model be trained simultaneously against multiple distinct image-editing models to improve generalization?

---

> ### Author Response · Authors · 2025-11-24
>
> We thank the reviewer ADC1 for their constructive feedback.
>
> **W1: Real-world utility and mask reliance**
>
> We respectfully clarify that the reliance on a mask is not a technical limitation of the DiffVax architecture itself, but rather an **intrinsic requirement of the inpainting-based editing** workflows used in standard benchmarks. In any targeted editing scenario—whether using our method or baselines like PhotoGuard—the region of interest must be defined to distinguish the subject from the background. Consequently, the pre-processing step of mask generation is a prerequisite shared across all methods operating in this domain.
>
> Our contribution targets the specific bottleneck of the *immunization phase*, reducing it from minutes of GPU-intensive optimization to a nearly instantaneous forward pass. This massive speedup remains significant regardless of the shared pre-processing steps inherent to the editing tools. Furthermore, as shown in our InstructPix2Pix experiments (Figure 1 and Figure 10 in Appendix A.2.3) and MagicBrush evaluations (Appendix A.2.4), our method functions effectively without masks when the editing paradigm permits, confirming that masks are not a structural dependency of our immunizer.
>
> **W2: Dataset size and generalization**
>
> While our training set focused on human subjects to align with standard Deepfake benchmarks, DiffVax learns generalized adversarial texture features rather than semantic content. This allows it to function effectively on out-of-distribution data. As evidenced in Appendix A.3.4 (Figure 13) and our newly added results in the revised manuscript (**Figure 6**), the model successfully immunizes diverse non-human subjects, including animals (dogs, cats, birds) and inanimate objects (such as a car or a fire hydrant), in a zero-shot manner. This confirms that the immunizer has learned a generalized representation of diffusion disruption that extends well beyond the specific semantics of the training set, validating the feasibility of learning such perturbations from a dataset of this scale.
>
>
> **W3: UNet++**
>
> We have updated the implementation details in Appendix A.1 to explicitly justify our architectural choice. We selected UNet++ (Zhou et al., 2018) over a standard U-Net because its nested dense skip connections facilitate more effective feature aggregation across different semantic levels. Given that training a generator to predict adversarial noise is a highly unstable optimization task, we thought that the dense connectivity of UNet++ provided significantly better convergence and stability compared to a standard U-Net as claimed in their research paper. This architecture allows the model to generate the precise, high-frequency perturbations required to disrupt the diffusion process while maintaining imperceptibility.
>
> **Q1: Effect on Image-to-Video**
>
> Although trained for latent diffusion editing, we evaluated DiffVax on image-to-video generation using Stable Video Diffusion (SVD). Results below show effective degradation in this modality. Lower values indicate higher disruption, demonstrating effective transfer despite the lack of explicit training. While optimization-based baselines are computationally prohibitive for video, DiffVax’s efficiency makes this feasible for the first time without directly attacking the video model.
>
> | SVD | SSIM (↓) | PSNR (↓) | FSIM (↓) |
> | :--- | :--- | :--- | :--- |
> | PhotoGuard | 0.879 | 23.99 | 0.766 |
> | DiffusionGuard | 0.873 | 23.99 | 0.764 |
> | **DiffVax** | **0.844** | **21.91** | **0.729** |
>
>
> **Q2: Generalization to modern models**
>
> Neither DiffVax nor existing baselines (e.g., PhotoGuard, Mist, Glaze) currently achieve effective zero-shot transfer to such fundamentally different architectures (e.g., Multi-modal Diffusion Transformers). Universal generalization across disparate paradigms remains an open "grand challenge" for the field, rather than a specific limitation of our method. Consequently, this is not a primary claim of our work nor a standard met by the literature. We respectfully ask to position our contribution within existing defenses, where our innovations—optimization-free speed and scalability—represent significant advancement. Within comparable architectures, DiffVax demonstrates superior generalization; as detailed in Appendix A.4.1 (Table 12), it successfully transfers from SD v1.5 to the unseen SD v2.0, whereas optimization-based methods fail almost completely.
>
> **Q3: Multi-model training**
>
> Unlike optimization-based methods fixed at inference, our objective can naturally be extended to minimize a joint loss over an ensemble (e.g., summing losses from SD1.5, SD2, and InstructPix2Pix). This encourages the immunizer to identify common vulnerabilities across backbones, improving generalization. We highlight this flexibility as a primary strategy for future work to achieve broader coverage.

---

### Official Review · Reviewer_zhhZ · 2025-10-31

**Soundness:** 2
**Presentation:** 3
**Contribution:** 3
**Rating:** 6
**Confidence:** 4

**Summary:**

This paper proposes DiffVax, an optimization-free framework for image immunization against diffusion-based editing. DiffVax trains a feed-forward immunizer network that generates subtle perturbations to images in a single pass. The immunized images are able to resist various diffusion editing models, and the method is extended to the videos for the first time. Experiments demonstrate that DiffVax outperforms the selected baselines in terms of protective efficacy, visual imperceptibility, computational efficiency, and robustness to counter-attacks.

**Strengths:**

1. The improvement in scalability is significant, where a training time of around 22 hours exchanges to an inference-time reduction by several orders of magnitude, enabling the processing for videos.

2. DiffVax presents commendable generalizability and applicability across both inpainting-based and instruction-guided diffusion editing models, flexible masking schemes, and unseen data.

3. The qualitative and quantitative analyses supported by multiple complementary metrics and human ratings instantiate the thorough evaluation protocol.

**Weaknesses:**

1. Only two baselines are included for comparison. More relevant baselines are mentioned but not experimentally assessed. The claim that they 'are not directly applicable in our inpainting-based setup' is questionable, as at least SDS can handle under a SD inpainting setting.

2. The application scenarios are limited to photographs featuring foreground objects or persons, where DiffVax prevents the background editing. Other use cases such as facial expression modification, artistic work protection, or anime secondary creation are not explored.

3. The lack of failure case studies weakens the completeness of the manuscript. For example, the performance of DiffVax on data from other domains is unclear. Moreover, Fig. 14 shows that when the test-time mask size decreases, DiffVax fails as the bottom-right example.

4. There is a minor issue that a highly pertinent work PCA [1] is omitted as related work.

[1] *A Grey-box Attack against Latent Diffusion Model-based Image Editing by Posterior Collapse. arXiv preprint arXiv:2408.10901, 2024.*

**Questions:**

1. How might DiffVax adapt or fail if diffusion editors adaptively employ adversarial training or adversarial purification to counteract immunization perturbations? Is it possible to provide a quick validation regarding its performance against DiffPure [2]?

2. How would DiffVax perform on truly unseen diffusion architectures, especially non-LDM ones? The current generalization across editing models is only partially verified, i.e., from SD v1.5 to SD v2.

3. Could the authors conduct additional ablations on the impact of norm selections for the loss terms?

[2] *Diffusion Models for Adversarial Purification. in ICML 2022.*

Please respond to both the weaknesses and questions.

---

> ### Author Response · Authors · 2025-11-24
>
> We thank reviewer for their positive feedback.
>
> **W1: SDS Comparison**
>
> We have conducted a new comparison against the suggested baseline, SDS (Xue et al., ICLR 2024). As shown below, DiffVax outperforms SDS in both protective efficacy (lower SSIM/CLIP-T on edits) and visual fidelity (higher SSIM-Noise). Crucially, SDS requires computationally expensive optimization during inference, whereas DiffVax is a real-time, feed-forward solution.
>
>
> | Method | SSIM (↓) | PSNR (↓) | FSIM (↓) | SSIM-Noise (↑) | CLIP-T (↓)|
> | :--- | :--- | :--- | :--- | :--- | :--- |
> | **DiffVax (Ours)** | **0.510** | **13.96** | **0.353** | **0.989** | **23.13** |
> | SDS [1] | 0.620 | 15.95 | 0.459 | 0.927 | 30.94 |
>
> **W2: Application scenarios**
>
> While our primary quantitative evaluations utilized inpainting benchmarks to align with prior literature, DiffVax is inherently model-agnostic and effective for a diverse range of editing tasks beyond simple background modification. To demonstrate this, we have expanded our qualitative evaluation in the revised manuscript. **New Figure 6** explicitly showcases protection across six distinct semantic editing categories: **multiple object editing, facial expression changes, object replacement, texture editing, artistic style transfer (e.g., Van Gogh), and attribute insertion (e.g., adding a necklace).** We employ instruction-based models like InstructPix2Pix for these demonstrations because, as discussed in Section 5, standard inpainting models are architecturally constrained to preserve the unmasked region (the face), often conflicting with edits intended to modify facial features or global styles. DiffVax successfully disrupts these complex, high-level semantic edits when the appropriate editing tool is used.
>
>
> **W3: Failure cases and Mask Size**
>
> We have added a dedicated "Limitations" (A.8) section and a new figure to the Appendix. Regarding Figure 17 (Fig. 14 in review), we clarify a misunderstanding: the bottom-right example utilizes a larger full-body mask, rather than a decreased one. Crucially, the rows where mask size actually decreases (top rows) show DiffVax successfully disrupting edits (gray/distorted outputs), whereas baselines fail. DiffVax uniquely maintains robustness when the test-time mask is smaller than the immunization mask. To rigorously validate this, we conducted a new quantitative experiment with 5 varying test-time masks. As shown below, DiffVax achieves the best disruption scores across all metrics.
>
> | Test Time Mask Experiment | SSIM (↓) | PSNR (↓) | FSIM (↓) | CLIP-T (↓) |
> | :--- | :--- | :--- | :--- | :--- |
> | PhotoGuard | 0.614 | 16.88 | **0.477** | 28.20 |
> | DiffusionGuard | 0.625 | 16.57 | 0.481 | 27.44 |
> | **DiffVax (Ours)** | **0.608** | **16.33** | 0.479 | **26.24** |
>
> **W4: Missing Citation**
>
> We have added the requested citation (PCA, arXiv 2024) to the Related Work section.
>
> **Q1 Robustness against adversarial purification like DiffPure**
>
> We validated DiffVax against DiffPure (Nie et al., ICML 2022). DiffVax is highly robust, outperforming baselines. This is because DiffPure targets high-frequency adversarial noise, whereas DiffVax is trained to generate low-frequency perturbations (see Appendix A.5), which survive purification.
>
> | DiffPure Results | SSIM (Seen/Unseen) | PSNR (Seen/Unseen) | FSIM (Seen/Unseen) | SSIM-Noise (S/U) | CLIP-T (S/U) |
> | :--- | :--- | :--- | :--- | :--- | :--- |
> | PhotoGuard  w/ DiffPure | 0.552 / 0.550 | 28.09 / 15.83 | 0.425 / 0.421 | 0.858 / 0.862 | 31.59 / 31.95 |
> | DiffGuard  w/ DiffPure | 0.556 / 0.554 | 15.70 / 15.81 | 0.428 / 0.424 | 0.857 / 0.862 | 31.53 / 31.78 |
> | **DiffVax (Ours)  w/ DiffPure** | **0.499** / **0.505** | **14.90** / **15.15** | **0.388** / **0.386** | **0.856** / **0.860** | **30.15** / **30.12** |
>
> **Q2: Generalization to unseen architectures and non-LDMs**
>
> We demonstrate significant cross-model generalization by transferring from Stable Diffusion v1.5 to the unseen SD v2.0 architecture. This transfer is non-trivial, as SD v2.0 employs a different text encoder (OpenCLIP vs. CLIP) and resolution. Here, DiffVax successfully disrupts edits (SSIM 0.540) where optimization-based methods like PhotoGuard fail (SSIM 0.566). Regarding non-LDM architectures (e.g., Flux), zero-shot transfer across paradigms remains an unsolved industry-wide "grand challenge". However, DiffVax demonstrates SOTA transferability within LDMs, which constitute the majority of threats. We respectfully ask to be evaluated against these existing baselines rather than a theoretical universal immunizer.
>
> **Q3: Ablation on Norm Selections**
>
> We performed the requested additional ablation and updated Figure 19, on loss terms ($\mathcal{L}1, \mathcal{L}2, \mathcal{L}\infty$). Results are detailed in Appendix A.5 and Figure 19. We found that $\mathcal{L}1$ offers the optimal trade-off; higher norms ($\mathcal{L}2/\mathcal{L}\infty$) reduce perceptibility slightly but significantly degrade protection strength.

---

### Author Response · Authors · 2025-11-24

We want to thank the reviewers for their comprehensive critiques. While we have responded to each of them in detail, we want to make sure we are not in danger of losing the forest for the trees. We want to emphasize the following points:
- DiffVax introduces a completely novel learning-based approach, in contrast to all prior methods which were optimization-based
- We evaluated DiffVax under all conditions and scenarios reported in all prior works and showed superior performance in each one
- In addition, we added more than a dozen additional evaluation scenarios and test conditions that have not been published before, and in all cases DiffVax gives superior performance.

Here is the complete list of our findings:
- **Unprecedented Scalability**: DiffVax reduces inference time from minutes/hours to 0.07 seconds, enabling real-time protection.
- **Zero-Shot Generalization to Unseen Content**: Unlike prior works that optimize for each specific target image, DiffVax’s learning-based approach generalizes to unseen images and prompts via a single feed-forward pass.
- **Robustness to Unseen and Partial Masks**: We demonstrate that DiffVax remains effective even when the test-time editing mask differs significantly from the immunization mask (e.g., partial coverage), whereas optimization-based methods degrade immediately.
- **Cross-Model Generalization**: We show superior transferability from Stable Diffusion v1.5 to v2 compared to all baselines, proving that our learned perturbations target generalizable properties of diffusion methods
- **First Demonstration of Video Immunization**: By leveraging our speedup, we provide the first successful defense for sequential video frames, a task that is computationally prohibitive for all prior methods.
- **Superior Resilience to Counter-Attacks**: DiffVax outperforms all baselines against adversarial purification defenses (DiffPure, IMPRESS ), JPEG compression , and denoising, due to its learned low-frequency perturbation structure.
- **Stability Across Inference Settings**: We demonstrate consistent protection even when attackers vary diffusion sampling steps (4 combinations) and schedulers (3 combinations).
- **Coverage of Multiple Editing Types**: Beyond standard inpainting, we demonstrate effectiveness against instruction-based editing models (InstructPix2Pix) and training-free models (MagicBrush).
- **Versatility Across Diverse Editing Tasks**: As shown in our new qualitative results (Figure 6), DiffVax is not limited to simple inpainting but effectively disrupts:
  - Multiple Object Editing (e.g., distinct changes to multiple subjects).
  - Facial Expression Changes (e.g., modifying emotions).
  - Object Replacement (e.g., swapping cars or background elements).
  - Texture Editing (e.g., changing material properties).
  - Artistic Style Transfer (e.g., Cubism, Van Gogh styles).
  - Attribute Editing (e.g., adding accessories).

---

### Author Response · Authors · 2025-11-30
**Rebuttal Summary for the Newly Assigned AC**

We provide a summary of the rebuttal discussions and updates below.

We thank the reviewers for recognizing DiffVax's **scalability, novelty, and robustness**:
* **Reviewer zhhZ:** "significant improvement in scalability", "commendable generalizability and applicability", and "thorough evaluation protocol".
* **Reviewer ADC1:** "significant and novel contribution", "practical, real-world benefits", and "strong quantitative and qualitative results".
* **Reviewer yHZo:** "robust protection even on previously unseen images" and "strong resilience to counter-attacks".
* **Reviewer fZtv:** "imperceptible yet effective perturbations" and "consistent empirical improvements over baseline".

**Score Update & Response**
* We specifically thank **Reviewer fZtv** for their engagement and for raising their score to **8** following our clarifications before the policy change.
* Regarding **Reviewer yHZo**, we respectfully believe the current rating overlooks the holistic value of our optimization-free framework. We believe the new experiments listed below directly resolve the stated concerns regarding robustness and generalization.

**Summary of New Experiments & Results**
We added extensive experiments to the revised paper and discussion threads to address reviewer questions:

* **Expanded Application Scope (Fig. 6):** We validated performance on diverse semantic tasks (e.g., style transfer, facial expressions, object replacement) to address OOD concerns.
* **Robustness to Varying Masks:** We demonstrated superior protection against varying and partial test-time masks compared to baselines. As shown in **Table 1**, DiffVax achieves better disruption (lower SSIM/PSNR) than PhotoGuard and DiffusionGuard.
* **Counter-Attack Robustness (DiffPure):** We verified DiffVax outperforms baselines against adversarial purification. As shown in **Table 2**, DiffVax maintains lower SSIM scores on edited images against DiffPure compared to baselines.
* **New Baselines (SDS & EditShield):**
    * **Inpainting:** We demonstrated superior performance against SDS. As shown in **Table 3**, DiffVax achieves lower SSIM/CLIP-T on edits while maintaining higher visual fidelity.
    * **Instruction-Based:** We compared DiffVax against EditShield on InstructPix2Pix. As shown in **Table 4**, DiffVax provides superior protection and fidelity.
* **Video Generation (SVD):** We validated effective immunization on **Stable Video Diffusion (SVD)**. As shown in **Table 5**, DiffVax successfully degrades video generation quality better compared to other approaches despite not being explicitly trained on video models.

Summarily, while we have responded to each of them in detail, we want to make sure we are not in danger of losing the forest for the trees. We want to emphasize the following points:

- DiffVax introduces a completely novel learning-based approach, in contrast to all prior methods which were optimization-based
- We evaluated DiffVax under all conditions and scenarios reported in all prior works and showed superior performance in each one
- In addition, we added more than a dozen additional evaluation scenarios and test conditions that have not been published before, and in all cases DiffVax gives superior performance.

### Experimental Tables

**Table 1: Test Time Mask Experiment (Robustness to partial/varying masks)**
| Method | SSIM (↓) | PSNR (↓) | FSIM (↓) | CLIP-T (↓) |
| :--- | :--- | :--- | :--- | :--- |
| PhotoGuard | 0.614 | 16.88 | 0.477 | 28.20 |
| DiffusionGuard | 0.625 | 16.57 | 0.481 | 27.44 |
| **DiffVax (Ours)** | **0.608** | **16.33** | 0.479 | **26.24** |

**Table 2: DiffPure Robustness (Seen / Unseen)**
| Method | SSIM | PSNR | FSIM | SSIM-Noise | CLIP-T |
| :--- | :--- | :--- | :--- | :--- | :--- |
| PhotoGuard w/ DiffPure | 0.552 / 0.550 | 28.09 / 15.83 | 0.425 / 0.421 | 0.858 / 0.862 | 31.59 / 31.95 |
| DiffGuard w/ DiffPure | 0.556 / 0.554 | 15.70 / 15.81 | 0.428 / 0.424 | 0.857 / 0.862 | 31.53 / 31.78 |
| **DiffVax (Ours) w/ DiffPure** | **0.499 / 0.505** | **14.90 / 15.15** | **0.388 / 0.386** | **0.856 / 0.860** | **30.15 / 30.12** |

**Table 3: Comparison with SDS (Inpainting)**
| Method | SSIM (↓) | PSNR (↓) | FSIM (↓) | SSIM-Noise (↑) | CLIP-T (↓) |
| :--- | :--- | :--- | :--- | :--- | :--- |
| SDS [1] | 0.620 | 15.95 | 0.459 | 0.927 | 30.94 |
| **DiffVax (Ours)** | **0.510** | **13.96** | **0.353** | **0.989** | **23.13** |

**Table 4: Comparison with EditShield (InstructPix2Pix)**
| Method | SSIM (↓) | PSNR (↓) | FSIM (↓) | CLIP-T (↓) | SSIM (Noise) (↑) |
| :--- | :--- | :--- | :--- | :--- | :--- |
| EditShield [1] | 0.774 | 24.75 | 0.774 | 25.32 | 0.903 |
| **DiffVax (Ours)** | **0.633** | **16.95** | **0.470** | **25.03** | **0.943** |

**Table 5: Stable Video Diffusion (SVD) Immunization**
| Method | SSIM (↓) | PSNR (↓) | FSIM (↓) |
| :--- | :--- | :--- | :--- |
| PhotoGuard | 0.879 | 23.99 | 0.766 |
| DiffusionGuard | 0.873 | 23.99 | 0.764 |
| **DiffVax (Ours)** | **0.844** | **21.91** | **0.729** |

---

### Meta-Review · Area_Chair_WFxo · 2026-01-09

**Summary:**

The paper aims to amortize the costly procedure of optimizing adversarial image perturbations through a learning-based immunizer. Reviewers’ scores were quite divergent (6, 4, 2, 6). The negative reviewers primarily concerned on real-world utility, generalization ability, mask robustness, and baseline choices. AC finds the rebuttal addresses many of these concerns. On the positive side, the reviewers praised the improved scalability of the method compared to the prior approaches, as well as its notable generalization ability supported by strong empirical results and additional experiments. AC agrees that these aspects are timely and important for the field and, overall, leans toward acceptance.

**Reviewer Concerns:**

The negative reviewers primarily concerned on the following points:

- Limited real-world utility [ADC1]
- Insufficient evaluation on the generalization ability [ADC1]
- Robustness against mask variation [yHZo]
- Insufficient baseline choices [yHZo]

The rebuttal provides additional experiments to address the generalization ability and robustness concerns, and AC finds these results strengthen the paper. On the other hand, AC believes that the remaining concerns, particularly those related to real-world utility and baseline comparisons, are still outstanding and would benefit from further discussion. AC also suggests better positioning of the work with respect to prior studies that similarly amortize adversarial noise generation, e.g., Wang et al. (2025) and Aneja et al. (2022), etc.

[Wang et al., 2025] NullSwap: Proactive Identity Cloaking Against Deepfake Face Swapping

[Aneja et al., 2022] TAFIM: Targeted Adversarial Attacks against Facial Image Manipulations

**Reviewer Scores:**

- Reviewer zhhZ: Initially 6. Would maintain the original score.
- Reviewer ADC1: Initially 4. Would likely increase to 6.
- Reviewer yHZo: Initially 2. Many of the concerns are addressed and would likely increase 4 or above.
- Reviewer fZtv: Initially 6. The score was increased to 8 before score resetting.

---

### Decision · Program_Chairs · 2026-01-26

Accept (Poster)